# Trade-offs between sperm viability and immune protein expression in honey bee queens (*Apis mellifera*)

Alison McAfee (ID) [1,2 ✉], Abigail Chapman (ID) [2], Jeffery S. Pettis[3], Leonard J. Foster[2] & David R. Tarpy[1]

Queens of many social hymenoptera keep sperm alive within their specialized storage organ, the spermatheca, for years, defying the typical trade-off between lifespan and reproduction. However, whether honey bee (*Apis mellifera*) queens experience a trade-off between reproduction and immunity is unknown, and the biochemical processes underlying sperm viability are poorly understood. Here, we survey quality metrics and viral loads of honey bee queens from nine genetic sources. Queens rated as 'failed' by beekeepers had lower sperm viability, fewer sperm, and higher levels of sacbrood virus and black queen cell virus. Quantitative proteomics on $N = 123$ spermathecal fluid samples shows, after accounting for sperm count, health status, and apiary effects, five spermathecal fluid proteins significantly correlating with sperm viability: odorant binding protein (OBP)14, lysozyme, serpin 88Ea, artichoke, and heat-shock protein (HSP)10. The significant negative correlation of lysozyme—a conserved immune effector—with sperm viability is consistent with a reproduction vs. immunity trade-off in honey bee queens.

[1] Department of Entomology and Plant Pathology, North Carolina State University, Raleigh, NC, USA. [2] Department of Biochemistry and Molecular Biology, Michael Smith Laboratories, University of British Columbia, Vancouver, BC, Canada. [3] Pettis and Associates LLC, Salisbury, MD, USA. ✉email: alison.n.mcafee@gmail.com

Long-term sperm storage is a remarkable feature of social insect biology, with hymenopteran queens storing sperm for by far the longest duration of any animal (decades)[1]. In some hymenopteran species, much work has been dedicated to sexual selection among males via sperm competition[2–6] and the trade-offs of male and female innate immunity with sperm quality and stored sperm viability[7–12]. However, the molecular processes linked to sperm viability during storage are not well understood. Honey bee queens could serve as an excellent model system to investigate such processes because they are highly amenable to empirical manipulation.

The reproduction versus immunity trade-off hypothesis—also known as the immunocompetence handicap—is a prevailing hypothesis in reproductive biology[7,13–16]. In males of a variety of species, including insects, there is a well-established negative relationship between sperm viability and immune function[8,10,12,17–21]. A similar trade-off appears to exist in female insects that engage in sperm storage[9,11,22].

These trade-offs between reproduction and immunity are thought to be driven either by resource-allocation compromises[23,24] or collateral damage of immune effectors[9,10]. The resource-allocation compromise states that the more biological resources a male or female invests in immune function, the lower the reproductive capacity (i.e., sperm quantity or quality in sperm-producing males[8,12], sperm storage in females[9,11,22], or ovum provisioning and production in females[7]). The alternate idea of collateral damage of immune effectors is based on the idea that sperm cells may be inadvertently damaged by innate immune defenses of the female[9]. In particular, collateral damage could occur via innate immune mechanisms that utilize bursts of cytotoxic reactive oxygen and nitrogen species[9,25]. Queen honey bees are under particularly strong selective pressure to minimize reactive oxygen and nitrogen species in the spermatheca in order to support long-term sperm maintenance. Indeed, the spermatheca is a largely anoxic environment[26], and mated queens upregulate enzymes that combat oxidative stress, like catalase and superoxide dismutase[27–29].

Both the collateral damage hypothesis and the resource allocation hypothesis predict that immunosuppressed individuals will have higher sperm viability, and likewise, that immune stimulation decreases sperm viability. This phenomenon has been observed in crickets[11] and fruit flies[9], where female immune stimulation (using peptidoglycan fragments) reduced sperm viability in the female's seminal receptacles. In addition, mating reduces phenoloxidase activity (an immune effector responsible for the melanization cascade) in wood ant and leaf-cutter ant queens[22,30].

In honey bee queens, the relationship between immune protein expression and sperm viability (whether via collateral damage or resource-allocation trade-offs) is, as yet, unexplored. Additionally, there is limited data on how proteins linked to sperm viability change after mating, when the queen must transition from storing no sperm to maintaining sperm viability for years. Here, we investigated the reproduction versus immunity trade-off hypothesis by performing quantitative proteomics on a large sample of genetically distinct queens, relating protein expression with stored sperm viability, and inspecting protein-protein correlation matrices for functional patterns.

## Results and discussion

**Evaluating quality metrics for healthy, failed, and imported queens.** For this survey, we initially sampled 125 queens belonging to three major cohorts: healthy queens ($n = 52$), failed queens ($n = 53$), and imported queens from commercial suppliers ($n = 20$, 10 each from California and Hawaii). Imported queens were treated separately from healthy queens because there was no prior assessment of quality before distribution. The sperm count and viability data did not satisfy all the assumptions for an analysis of variance (ANOVA) test, therefore least squares and weighted least squares linear models were used where appropriate (Table 1). We also included queen producer as a fixed effect to account for potential genetic or environmental differences between sources. The average sperm viability and sperm counts were nearly identical between healthy queens and imported queens, but failed queens had

**Table 1 Statistical parameters.**

| Factor | Level | Shapiro p | Levene p | Statistical method | Contrasts | | |
|---|---|---|---|---|---|---|---|
| | | | | | Group[a] | p | \|t\| |
| Sperm viability[b] | Healthy | 0.0534 | 0.000146 | Weighted least-squares linear model | H–F | 0.00132 | 3.29 |
| | Failed | 0.182 | | | F–I | 0.181 | 1.34 |
| | Imported | 0.114 | | | I–H | 0.94 | 0.075 |
| Sperm counts[b] | Healthy | 0.160 | 0.0714 | Least squares linear model | H–F | 0.00472 | 4.86 |
| | Failed | 0.630 | | | F–I | 0.0245 | 2.28 |
| | Imported | 0.250 | | | I–H | 0.448 | 0.762 |
| Ovary mass[b] | Healthy | 0.00774 | 0.0506 | Least squares linear model | H–F | 0.395 | 0.855 |
| | Failed | 0.3279 | | | F–I | 0.36 | 0.920 |
| | Imported | 0.0919 | | | I–H | 0.6545 | 0.449 |
| DWV[c] | Healthy | $2.25 \times 10^{-7}$ | 0.000436 | Weighted least squares linear model | H–F | 0.000485 | 3.61 |
| | Failed | $3.92 \times 10^{-12}$ | | | F–I | 0.00824 | 2.70 |
| | Imported | 0.00460 | | | I–H | 0.436 | 0.781 |
| SBV[c] | Healthy | NA | $4.72 \times 10^{-16}$ | Weighted least squares linear model | H–F | 0.000296 | 3.75 |
| | Failed | $8.84 \times 10^{-7}$ | | | F–I | 0.000107 | 4.04 |
| | Imported | NA | | | I–H | 0.138 | 1.50 |
| BQCV[c] | Healthy | $2.70 \times 10^{-8}$ | 0.149 | Least squares linear model | H–F | $1.66 \times 10^{-5}$ | 4.53 |
| | Failed | 0.0019 | | | F–I | $9.3 \times 10^{-5}$ | 4.08 |
| | Imported | $3.12 \times 10^{-5}$ | | | I–H | 0.402 | 0.826 |
| Total virus[c] | Healthy | $3.81 \times 10^{-5}$ | 0.0245 | Weighted least squares linear model | H–F | 0.0769 | 1.79 |
| | Failed | 0.000357 | | | F–I | 0.291 | 1.06 |
| | Imported | 0.00459 | | | I–H | 0.779 | 0.282 |

[a]H = Healthy, I = Imported, F = Failed
[b]Queen producer was included as a fixed effect
[c]Producer could not be included as a fix effect due to singularities

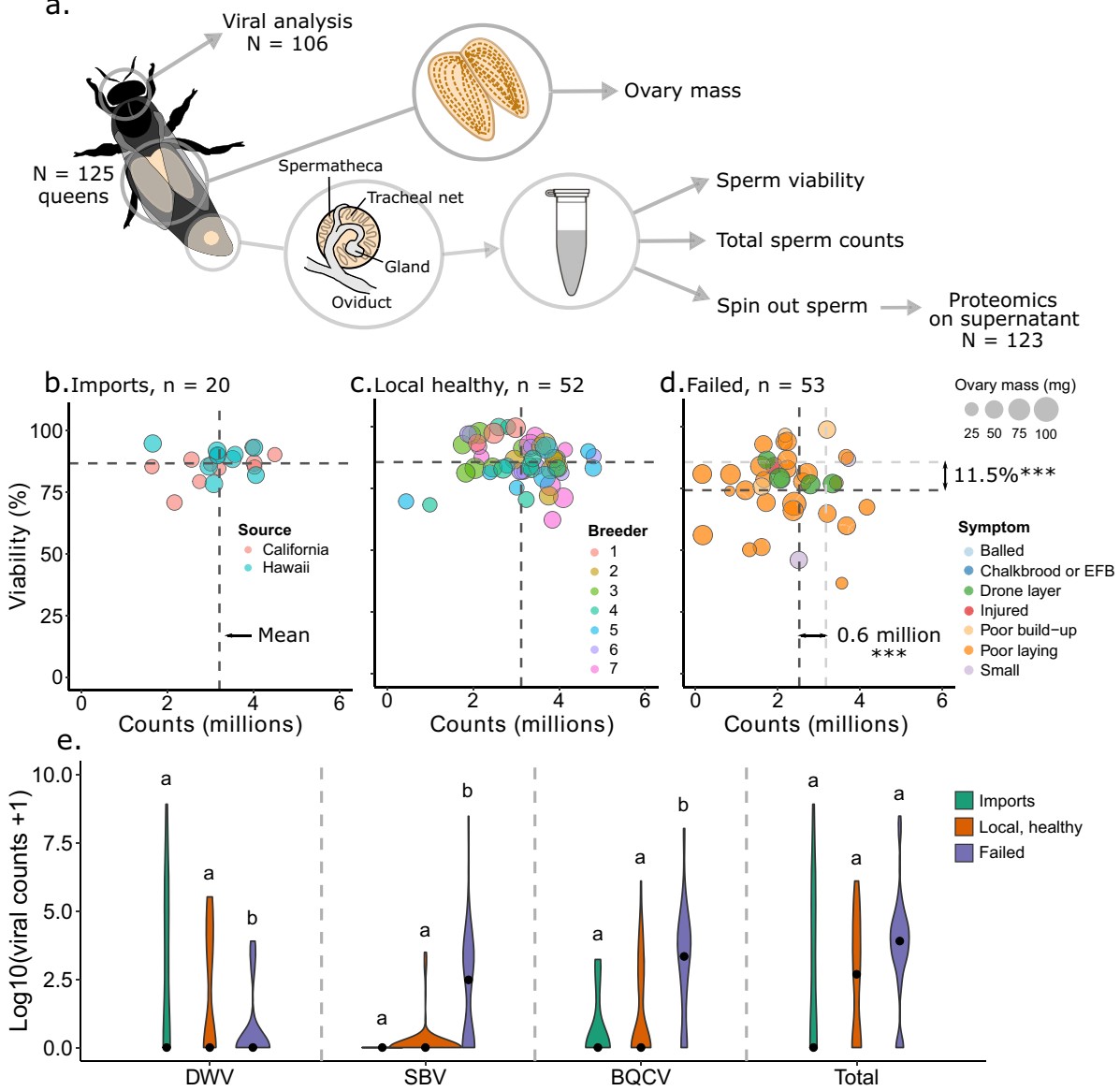

**Fig. 1 Sperm viability, sperm counts, ovary mass, and viral metrics.** See Table 1 for a complete summary of statistical tests, parameters, and p values. **A** Experimental schematic. **B** imported, **C** healthy, and **D** failed queens were surveyed for macroscopic health metrics. Symptoms shown are as reported by donating beekeepers. Sperm viability and sperm count data were acquired for 125 queens, while 123 queens contributed to final proteomics data owing to sample loss during handling. *** indicates $p < 0.005$. (**E**) For a subset of the queens, viral copy numbers were measured using RT-qPCR ($n = 44$ healthy queens, $n = 13$ imported queens, and $n = 49$ failed queens). Lower-case letters indicate statistical significance ($p < 0.05$) within each virus.

significantly lower sperm viability ($p = 0.00132$, t = 3.29, df = 118) and sperm counts ($p = 0.00472$, t = 4.86, df = 117) compared to healthy queens (Fig. 1a–d). Ovary masses also differed significantly between groups – imported queens had significantly lower ovary masses compared to either healthy queens ($p = 0.0021$, t = 3.1, df = 134) or to failed queens ($p = 0.038$, t = 2.1, df = 134), but ovary mass can be strongly influenced by caging time and worker care. These differences are not detectable when "producer" is included as a fixed effect in the statistical model, since all the imported queens were produced by California and Hawaiian suppliers. Among the local and imported queens, sperm viability was consistently high across all producers and import sources, and sperm counts were statistically similar (Supplementary Figure 1). Sample metadata is available in Supplementary Data 1.

Since the healthy and failed queens are not age-matched, we cannot say if the differences in counts and viability that we

observed are due to age or quality differences. Furthermore, in preliminary analyses we found that the ovary-mass data only marginally passed Levene's test for equal variance ($p = 0.051$), which was driven by low variation in imported queen ovary masses. This, on top of imported queens having the smallest ovaries, strongly suggests that the data are not from the same statistical population. In follow-up experiments, we observed that the ovary mass of imported queens is regained after two weeks spent caged inside a colony, and therefore is not likely an intrinsic quality of imported queens (Supplementary Figure 2). Rather, it is likely an artifact of longer caging duration during international transit.

**Viral analysis.** To assess patterns of viral abundance in the queen cohorts, we measured deformed wing virus (DWV), sacbrood

virus (SBV), and black queen cell virus (BQCV) levels. We first analyzed a subset of 45 queens for DWV, SBV, BQCV, as well as acute bee paralysis virus, Kashmir bee virus, and Israeli acute paralysis virus, but found no detectable levels of the latter three. We, therefore, analyzed a further 61 queens for only DWV, SBV, and BQCV (in total, $n = 44$ healthy queens, $n = 13$ imported queens, and $n = 49$ failed queens). We found that failed queens had significantly higher copy numbers of SBV and BQCV relative to imported queens and healthy queens but lower copy numbers of DWV (Fig. 1e). Combining copy numbers of all three viruses into a total viral load, failed queens had higher, but not statistically significant ($p = 0.077$, $t = 1.79$) loads than healthy queens (see Table 1 for all associated p values). We also identified significant effects of queen source (producer) for all three viruses, indicating that the apiaries from which the queens came had characteristic viral profiles (DWV: $p = 4.66 \times 10^{-6}$, F = 5.2, df = 11 and 94; SBV: $p = 3.95 \times 10^{-8}$, F = 6.66, df = 11 and 94; BQCV: $p = 0.0359$, F = 2.01, df = 11 and 94; Total load: $p = 2.44 \times 10^{-8}$, F = 6.83, df = 11 and 94). N = 94 queens had both viability and virus data, among which sperm viability was not dependent on viral copies (DWV: $p = 0.330$, $t = -0.979$; SBV: $p = 0.424$, $t = 0.802$; BQCV: $p = 0.579$, $t = 1.79$; Total: $p = 0.878$, $t = -0.153$). Both the viral and sperm viability data were highly variable; however, failed queens tended to have higher viral loads as well as lower sperm viability, which is consistent with a reproduction-immunity trade-off.

**Proteomics analysis on spermathecal fluid.** We first took a broad view of proteins linked to sperm viability by correlating protein expression in the spermathecal fluid to the viability of stored sperm (underlying proteomics data and statistics are available in Supplementary Data 2 and 3). Of the 2,512 proteins identified (1% false discovery rate based on reverse hits) and 1,999 quantified (proteins identified in fewer than 10 samples were removed), five specific proteins significantly correlated with sperm viability at a 10% false discovery rate (Benjamini-Hochberg method): Lysozyme, Odorant binding protein (OBP)14, Serpin 88Ea, Artichoke, and Heat-shock protein (HSP)10 (Fig. 2a, Table 2). Since queen source (producer) was included as a random effect in our statistical model, these differences are unlikely to be a result of source bias. Furthermore, colony health status ('failed,' 'healthy,' and 'imported') was included as a fixed effect in the model, and since queens heading failed colonies also tended to be older and had a higher viral load, these proteins are unlikely to simply be linked to sperm viability indirectly through aging queens or differences in viral titer (queen ages, where known, are listed in Supplementary Data 1). To be sure, we checked if the expression levels of these five specific proteins were linked to viral copy numbers (individual viruses as well as total load) using N = 94 queens with complete virus and viability data, and found no significant relationships (Supplementary Figure 3; least squares linear model, including sperm viability, queen status (levels: healthy, failed, imported), and viral copy numbers for DWV, SBV, BQCV, and Total load as fixed effects). No significant relationships were identified ($p > 0.05$) except with sperm viability, which we already determined in the original proteome analysis ($p < 0.005$). See Supplementary Figure 3 for complete statistical reporting.

Lysozyme is a well-known immune effector that is negatively related to sperm viability in multiple cricket species[8,12,31]— although not entirely unequivocal, this result is consistent with the notion that reproduction versus immunity trade-offs may exist in honey bee queens. However, we cannot exclude that natural infections could be impacting both immune protein expression and quality metrics. While DWV, SBV, and BQCV were detectable,

these viruses were not linked to the expression of the top proteins linked to sperm viability. However, this is not an exhaustive list of potential pathogens. It is also possible that immune proteins could be elevated as a consequence of sperm death, rather than preceding it. However, in other experiments, we have experimentally stressed queens using techniques that are known to reduce stored sperm viability (i.e. heat exposure)[29] and we did not observe elevated levels of any of the significant proteins we identified here.

Using the gene score resampling approach, several gene ontology (GO) terms were significantly enriched among proteins correlating with sperm viability (10% false discovery rate, Benjamini-Hochberg method), with odorant binding being one of the most significant after correction for protein multifunctionality and multiple hypothesis testing (Fig. 2b). Furthermore, OBP14 is strongly upregulated in mated queens relative to virgin queens and has relatively low abundance in semen (Fig. 2c), suggesting that its upregulation may be controlled by the act of mating or the presence of sperm. The strong, significant enrichment of odorant-binding is driven by the combined effect of the aforementioned OBP14 correlation, as well as weaker correlations of OBPs that are co-expressed with OBP14 (OBP3, 4, 13, 16, 19, and 21), all of which are negatively (but not significantly) correlated with sperm viability (Fig. 2d). The diversity of OBPs in drone ejaculates (Fig. 2e) is consistent with previous reports of odorant reception regulating sperm motility[32], but the significantly higher expression of OBP14 in the spermatheca relative to ejaculates, and its negative correlation with viability, suggest an alternate role in the context of sperm storage.

Although the abundance of OBP14 in semen is low, it is possible that sperm death and subsequent release of proteins could contribute to the abundance patterns we observe (the same is true for Serpin 88Ea, which is also present in semen). To check this, we correlated protamine-like protein (a highly abundant sperm nuclear protein)[33] with both sperm viability and absolute number of dead sperm, and found no significant correlations (Supplementary Figure 4, Pearson correlation, $r = 0.218$, $p = 0.215$ and $r = -0.176$, $p = 0.321$, respectively). The protein was also sparsely identified in only 36 out of 123 samples, and was likely a result of sporadic sperm lysis during sample handling. Therefore, we reason that it is unlikely that the release of sperm proteins upon death can explain the negative correlations we observe for OBP14, Serpin 88Ea, Lysozyme, and Artichoke.

Like other OBPs, OBP14 is a soluble, globular protein with a hydrophobic ligand-binding pocket. OBP14 is the only honey bee OBP with a crystal structure, and previous work suggests that it preferentially binds terpenoid molecules[34,35]. Juvenile hormone (JH) is a sesquiterpenoid insect hormone with numerous functions related to development, immunity, and reproduction[36–38], making it an appealing candidate ligand for OBP14. JH has immunosuppressive effects in specific contexts;[39,40] therefore, it has the features of a key mediator for controlling the reproduction-immunity trade-off. We thus speculate that in the spermathecal fluid, OBP14 may be involved in hormonal signaling that regulates queen immunity, and OBP14-mediated JH signaling may influence sperm viability directly or indirectly through immune effects. We reason that if OBP14 were to bind and sequester free JH, JH may be less able to exhibit its immunosuppressive effects, thus lowering sperm viability by tipping the reproduction-immunity trade-off in favor of immunity. Alternatively, the proposed OBP14-JH complex may bind specific receptors and initiate physiological changes through signaling, rather than sequestration. Further experiments will be necessary to determine the specific molecular mechanisms involved.

HSP10, which positively correlates with sperm viability, is a protein chaperone expressed in the mitochondria, but it is also released into extracellular fluid[41]. In vertebrates, it is a negative regulator of immunity—indeed, it is also known as the "early

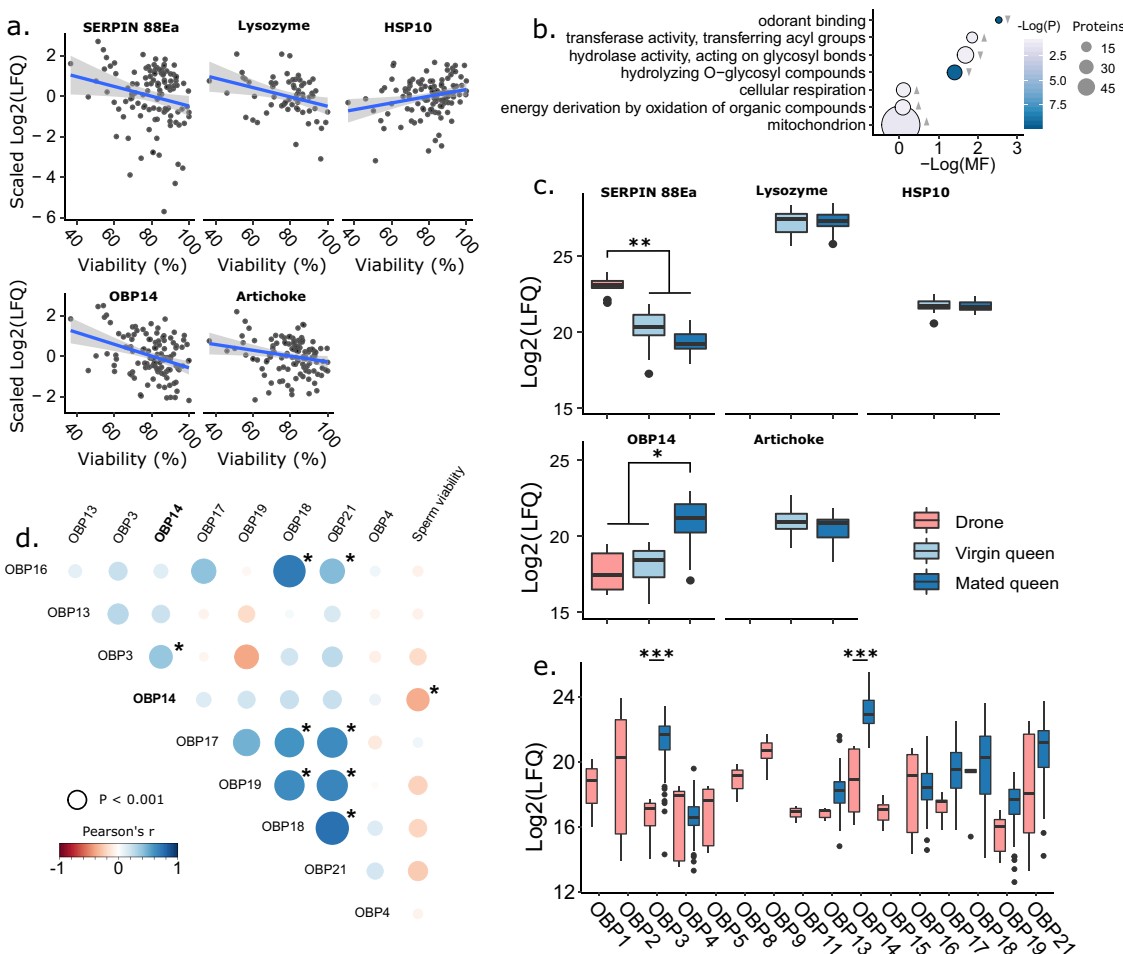

**Fig. 2 Proteins associated with sperm viability. a** We analyzed the spermathecal fluid of $N = 123$ queens by label-free quantitative proteomics. The linear model included sperm viability, sperm counts, and cohort (healthy, failed, and imported) as fixed factors, queen producer as a random effect, and the false discovery rate (FDR) was set to 10% (Benjamini-Hochberg method). The y axis depicts mean-centered label-free quantification intensity data after log2 transformation. Adjusted p values are shown. Statistical parameters can be found in Supplementary Data 3. Shaded gray bands represent the 95% confidence interval. **b** Gene ontology (GO) terms that are significantly enriched among proteins correlating with sperm viability using the gene score resampling method. **c** Protein expression in drone ejaculates, virgin spermathecae, and mated spermathecae (each with N = 10). Data were analyzed using an analysis of variance followed by Tukey contrasts. Serpin 88Ea contrasts: Drone-Virgin $p < 1 \times 10^{-7}$, Drone-Mated $p < 1 \times 10^{-7}$. OBP14 contrasts: Mated-Drone $p = 3.9 \times 10^{-5}$, Mated-Virgin $p = 8.0 \times 10^{-5}$. **d** Protein–protein and protein-viability correlation matrix of odorant-binding proteins in the spermathecal fluid. Dot size is proportional to significance. Significant correlations are indicated with an asterisk ($\alpha = 0.0011$, Bonferroni correction). **e** Odorant binding protein (OBP) expression in drone ejaculates and mated queens ($N = 10$ each). Data underlying panels **c** and **e** were previously published[29]. Data were analyzed by a two-way analysis of variance, which indicated an interactive effect between sex and protein (df = 8, F = 8.7, $p < 1.8 \times 10^{-11}$) followed by Tukey contrasts. OBP3 and OBP14 contrasts: $p < 1.0 \times 10^{-7}$. In all cases, boxes represent the bounds between the 2nd and 3rd interquartile range, midlines represent the median, and whiskers are extended by 1.5 times the interquartile range.

---

**Table 2 Functional description of proteins significantly correlating with sperm viability.**

| Protein description | Accession | General function(s) | Viability correlation |
|---|---|---|---|
| Odorant binding protein 14 | NP_001035313.1 | Solubilization of semiochemicals, ligand transport, preferential binding to terpenoid molecules[34,35] | Negative |
| Artichoke | XP_026295178.1 | Essential for cilia and flagella beating in *Drosophila*[47] | Negative |
| Serine protease inhibitor 88Ea | XP_026298978.1 | Negative regulator of Toll[54] | Negative |
| Lysozyme | XP_026300526.1 | Antibacterial and antifungal activity, associated with low sperm viability in crickets[8,12,31] | Negative |
| 10 kDa heat-shock protein (HSP10) | XP_624910.1 | Constitutive protein chaperone | Positive |

---

pregnancy factor" because its expression facilitates zygote implantation in the uterus via immunosuppression in the mother (likely through interactions with mammalian Toll-like receptors)[42]. This idea is analogous to the collateral damage of immune effectors on

stored sperm observed in *Drosophila*[9]. An immunosuppression function of HSP10 has not been demonstrated in invertebrates, and its function in insects, apart from its role as a chaperone, has received little attention[43–45].

**Protein-protein co-expression matrices.** The proteins we identified as correlating with sperm viability are multifunctional and, in some cases, poorly characterized. We, therefore, exploited protein correlation matrices and hierarchical clustering to make further inferences about the proteins' potential functions based on proximal associations with other proteins—an approach that has been widely used in other systems[46]. The guiding principle is that co-expressed proteins are more likely to function in the same biochemical pathway, or be physically interacting as components of a protein complex[46].

We computed Pearson correlation matrices including all 1,999 quantified proteins, then performed hierarchical clustering to

group those proteins that are co-expressed (Fig. 3a–d, Supplementary Data 4). We first confirmed that the clusters we defined are biologically meaningful by testing for GO terms enriched within each protein cluster. Of the 79 clusters we defined (to which 1,377 proteins belong, singletons and doubletons removed), 18 of them had enriched GO terms for biological functions or molecular processes, demonstrating that these are likely to be biologically meaningful groupings. Next, we identified the clusters to which our five proteins of interest belong, and found that Serpin 88Ea, Lysozyme, and Artichoke are part of the same cluster (cluster 5), in addition to numerous proteins involved in pathogen-associated molecular pattern recognition

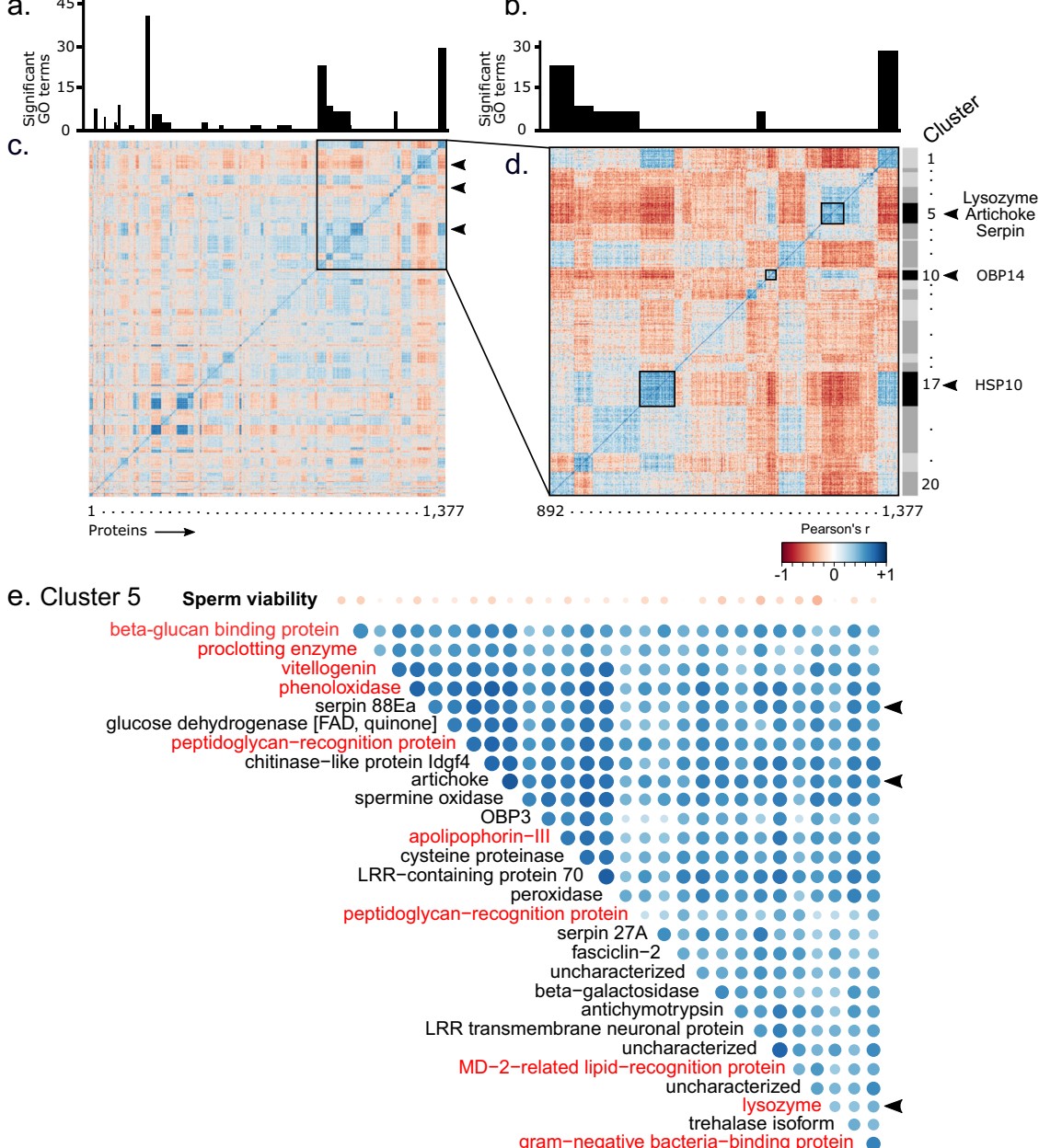

**Fig. 3 Protein–protein co-expression matrices.** We computed protein-protein Pearson correlations across $n = 123$ queen spermathecal fluid samples at a dendrogram cut-off of 666 clusters (one third of the total number of quantified proteins). Singleton clusters were removed before plotting. (**a**) Significantly enriched GO terms within clusters (Fisher exact test, 10% false discovery rate, Benjamini-Hochberg method). (**b**) GO terms of inset clusters. (**c**) All protein clusters (using a cut-off of k = 666 clusters, singletons and doubletons removed). (**d**) Inset protein clusters containing the five proteins correlating with sperm viability. (**e**) All protein members of cluster 5 and their relationship with sperm viability. Protein names in red indicate proteins directly involved in innate immune signaling.

and others involved in cellular encapsulation immune reactions (Phenoloxidase, Proclotting enzyme, and Apolipophorin-III; Fig. 3e and Supplementary Data 5). Twelve of the 27 proteins belonging to Cluster 5 are linked to innate immunity. All of these innate immune factors are also negatively (but not significantly, individually) correlated with sperm viability. On top of the significant negative correlation of Lysozyme expression with sperm viability, these data are consistent with the idea that queens are subject to reproduction-immunity trade-offs when it comes to sperm storage.

Very little is known about the artichoke protein's function, but some evidence demonstrated that it is essential for cilia and flagellar function[47]. In the quiescent state of the sperm during storage, flagellar motion is undesirable because it demands large amounts of ATP and is not necessary for sperm storage. This is consistent with the negative relationship we observe between Artichoke and viability. However, if this were its primary function, we would expect artichoke to also be present in drone ejaculates, which we did not observe (Fig. 2c). Artichoke has not been previously linked to immunity, but it is an understudied protein; because it clusters with known immune effectors and regulators here, that is one alternate role that should be explored.

While the cluster containing OBP14 did not yield any significant GO terms, two of the other cluster members are Apolipophorin I/II and Hexamerin 70a, both of which are also involved in JH binding in other insects[48–50], suggesting that OBP14, Apolipophorin I/II, and Hexamerin 70a could be facilitating hormone trafficking. Others have shown that JH diet supplementation improves sperm viability[51], and JH serves as an immunosuppressant in mated females of other insects, which is consistent with the reproduction-immunity trade-off hypothesis[52]. Indeed, Kim et al[53]. recently identified a mosquito OBP which binds JH and activates innate immune defenses—a mechanism which, according to the reproduction-immunity trade-off hypothesis, would be consistent with high levels of OBP14 being associated with low sperm viability in our data.

**No evidence for Serpin 88Ea inhibitory activity**. In *Drosophila*, Serpin 88Ea is a negative regulator of Toll immune signaling, and here it clusters with other proteins linked to innate immunity. However, the negative correlation with sperm viability and positive correlation with downstream immune effectors is not consistent with the reproduction-immunity trade-off hypothesis, nor this immune regulatory role. If the reproduction-immunity trade-off applies here and Serpin 88Ea functions in honey bees as it does in fruit flies, Serpin 88Ea should be positively correlated with sperm viability and inhibit expression of immune effectors. In *Drosophila*, Serpin 88Ea regulates Toll signaling by blocking proteolytic cleavage of Spaetzle by Spaetzle-processing enzyme, which is a necessary step for Toll activation[54]. Despite Serpin 88Ea levels positively correlating with both Spaetzle and Spaetzle-processing enzyme (Fig. 4a–c), we find no support that Serpin 88Ea is actually inhibiting Spaetzle-processing enzyme in our data. Serpins inhibit proteases by forming a covalent bond with the protease at its active site and inducing a conformational change[55], so if Serpin 88Ea is predominantly functioning as a protease inhibitor here, this should be confirmed in the mass spectrometry data. It would be unlikely that we would have identified the serpin-protease linkage because such bridged peptides fragment unpredictably in the mass spectrometer and non-canonical covalent bonds are not accounted for in the protein search database. However, we should be able to see the absence or decreased abundance of a peptide, either from the serpin or the protease, if this linkage is occurring. Unfortunately, it would not

be possible to see a cleaved spaetzle peptide in our data because spaetzle cleavage occurs C-terminal to an arginine residue (Fig. 4d), which would be indistinguishable from a cleavage by trypsin, the enzyme used in our sample preparation.

The honey bee homolog of Serpin 88Ea has not been characterized, so we used Basic Local Alignment Search Tool to identify the conserved reactive center loop region and confirmed that the protein contains the consensus sequence characteristic of inhibitory serpins (Fig. 4e). In *Drosophila*, the site targeted for nucleophilic attack by the protease occurs between residues 386 and 387 (TYR**S/A**RPV) in the reactive center loop. Therefore, the predicted cleavage site in honey bee Serpin 88Ea is between residues 369 and 370 (TFR**S/G**RPL), which is contained in the tryptic peptide SGRPLVPTVFNANHPFVYFIYEK (the 'site peptide'). We evaluated intensities of two control peptides (distant from the nucleophilic attack site) and the site peptide relative to a fourth reference peptide. All peptides were tightly correlated to the reference peptide at approximately the same slope (Fig. 4f–g), suggesting that intensities of the site peptide were not decoupled and that in this biological context Serpin 88Ea is not actively inhibiting proteases.

However, we acknowledge that this is an imperfect analysis, since we do not have a good positive control serpin (one which, under our experimental conditions, is known to appreciably covalently bind a protease). Therefore, we cannot confirm the degree of site peptide decoupling that we should expect if the serpin is acting as an inhibitor. In the future, we aim to conduct experiments involving spiking spermathecal fluid with increasing doses of a serine protease to confirm the expected concomitant decoupling of the site peptide from alternate peptides for an array of predicted inhibitory serpins.

Curiously, Dosselli et al. recently found that a different Serpin, B10 (along with two serine proteases, Easter and Snake), became downregulated in the ant *Atta colombica* seminal fluid upon exposure to spermathecal fluid[56]. The authors suggest that the proteases and Serpin B10 are part of a sperm-sperm competition system that becomes quickly deactivated by spermathecal fluid to preserve sperm viability. In our data, Serpin 88Ea along with two other protease inhibitors, Serpin 27 A (which targets Easter) and Antichymotrypsin, are all negatively associated with sperm viability. While this is consistent with the overall de-activation of sperm competition favoring viability, it is hard to rationalize how that could be the case here, since honey bee drone sperm and not the seminal fluid migrate through the queen's reproductive tract to the spermatheca. Therefore, if these proteins are the remains of sperm-competition machinery, they would have had to enter the spermatheca by physically associating with the membranes of sperm cells—a highly unlikely scenario, especially given that our statistical model yielded no proteins correlating with sperm counts.

Interestingly, in mammals, some serpins actually act as hormone carriers[57]. Although these are typically 'non-inhibitory' serpins without a reactive center loop domain and the serpins in our data are ostensibly inhibitory, it is possible that insect serpins have evolved diverse roles. Indeed, despite containing a reactive center loop domain, we found no evidence that Serpin 88Ea is actively serving as a serine protease inhibitor. In addition, the top predicted protein interactor for *Drosophila* Serpin 88Ea, according to STRING (a database of protein-protein interactions; www.string-db.org), is actually an apolipophorin involved in JH transport (FBpp0088252, score: 0.919). Further experiments will be necessary to ascertain whether Serpin 88Ea is associating with hormone carriers, is an immune regulator, is a component of sperm competition machinery, or some combination of different functions in different biological contexts.

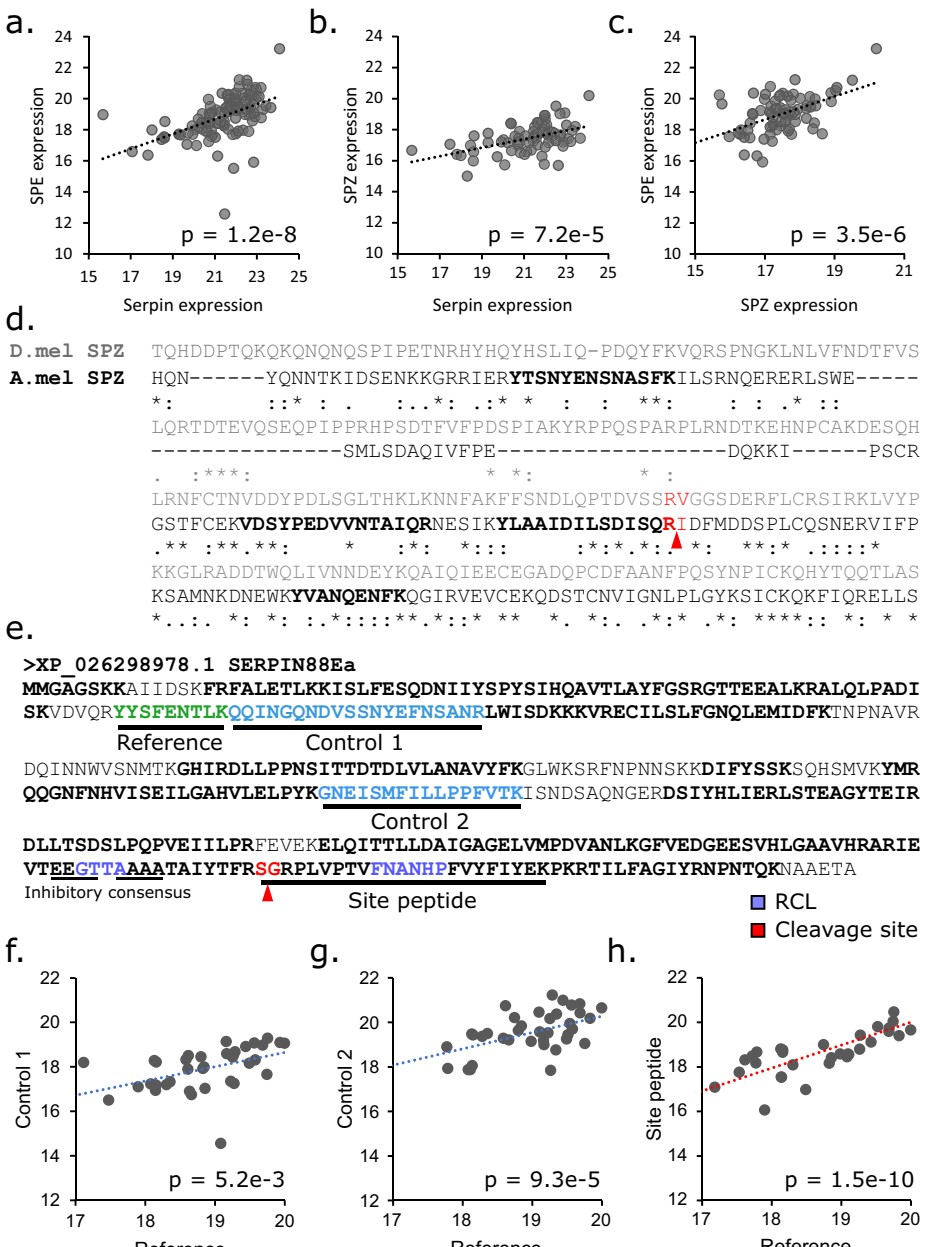

**Fig. 4 An investigation into Serpin 88Ea, Spaetzle (Spz), and Spaetzle-processing enzyme (SPE).** The three proteins are all positively, significantly correlated with each other (linear model on log2 transformed LFQ intensities). **a** SPZ-Serpin: N = 114, degrees of freedom (df): 113, F = 37.9, $p = 1.2 \times 10^{-8}$. **b** N = 75, df = 74, F = 17.8, $p = 7.2 \times 10^{-5}$. **c** N = 75, df = 74, F = 25.2, $p = 3.5 \times 10^{-6}$. **d** Sequence alignment between *Drosophila* (D.mel) Spz and honey bee (A.mel) Spz (* = perfect homology, : = good homology, . = partial homology, based on BLOSUM matrices). Peptides identified by mass spectrometry are bold. The predicted Spaetzle cleavage site is marked in red. **e**) Annotation of honey bee Serpin 88Ea sequence with predicted cleavage site (red), reactive center loop (RCL, purple), and inhibitory consensus sequence (thin underline). The site peptide contains an internal arginine residue (R); however, it is followed by a proline residue and therefore is not predicted to be cleaved by trypsin. **f–h** LFQ intensities of the site peptide and control peptides are compared to the reference peptide. **f** N = 35, df = 34, p = 0.0052; F = 8.9. **g** N = 37, df = 36, p = 0.0093; F = 18.9. **g**) N = 29, df = 28, $p = 1.5 \times 10^{-10}$; F = 81.7.

## Methods

**Queens**. Seven queen producers throughout BC (located in Grand Forks, Merritt, Armstrong, Abbotsford, Telkwa, Surrey, and Powell River) donated healthy queens for this study in the summer of 2019. All queens were approximately two months old and rated as "good quality" by the donors based on having a consistent, contiguous laying pattern. The sperm viability metrics for failed and healthy queens are the same results as described in McAfee et al;[29] however, all sperm count, ovary mass, and imported queen data are novel. The queens were part of a regional survey of participating operations: Queens from different operations were handled similarly and not exposed to environmental stressors in the laboratory. Two of the same producers in Grand Forks also donated the majority of failed queens, but

other donors located in Squamish, Abbotsford, Cranbrook, Lillooet, and Vancouver also contributed. In most cases, the exact age of the failed queens was unknown. Queen years and approximate ages (in months) are listed in Supplementary Data 1.

The queens arrived via overnight ground transportation (ACE Courier or via post) to the University of British Columbia in Vancouver and were sacrificed for analysis immediately upon arrival. Imported queens were shipped from producers in Hawaii and California to Edmonton, Alberta, then shipped together to Vancouver within hours of arrival. The queens arrived at 10:30 pm and were dissected and analyzed at the University of British Columbia on the following morning. Queens across all sources were inspected for *Nosema* spores using

standard microscopy methods[58]. See Supplementary Data 1 for complete sample metadata. As non-cephalopod invertebrates, honey bees are not subject to animal ethics approval at UBC.

**Measuring sperm viability, sperm counts, and ovary masses**. We conducted sperm viability assays exactly as previously described[29], with the original method published by Collins et al[59]. Briefly, we dissected spermathecae and lysed them in tubes containing 100 µl of room-temperature Buffer D. We transferred 10 µl of the solution to a new tube and stained it with propidium iodide and Sybr green fluorescent dyes, which differentially stains dead sperm red and live sperm green. After incubating for 15 min, we acquired images by fluorescent microscopy (three fields of view per queen) and sperm belonging to red and green channels were automatically counted using ImageJ. Sperm that stained both green and red were counted as live, as they were likely in the process of dying as a result of dissection or associated extraneous variables. We used ImageJ version 1.52a to count sperm cells and we averaged the percent viability across the three technical replicates prior to performing protein correlations.

We counted total sperm using the methods essentially as described by Baer et al[60]. Briefly, we mixed the sperm suspended in Buffer D solution by gently flicking the tube several times until homogeneous, then pipetted three 1 µl spots on to a glass slide, allowing it to air dry for 20 min. We stained the spots with DAPI, then imaged the entire area of each spot with a fluorescent microscope (images were taken at 200x in a tiling array, then automatically stitched together in the Zeiss image processing software, ZEN). The number of sperm nuclei in each 1 µl spot was then counted in ImageJ and averaged across the three spots. We arrived at the total number of sperm by extrapolation (multiplying the average value by a factor of 100). Finally, we dissected ovaries from the queens using forceps and wet weights were determined using an analytical balance, subtracting the exact mass of the tube. The remaining queen tissue was stored at −70 °C until further analysis.

**Proteomics sample preparation**. For each queen, we used the remaining ~87 µl of spermathecal solution that was not consumed by the viability and count assays for shot-gun proteomics. We removed spermathecal wall debris and sperm cells by centrifugation (1000 g, 5 min), transferring the supernatant to a new tube. The proteomics samples, therefore, were composed only of spermathecal fluid. We diluted the samples 1:1 with distilled water, then precipitated the proteins by adding four volumes of ice cold 100% acetone and incubating at −20 °C overnight. The final sample count for proteomics was 123 out of 125 initial queens owing to sample losses during handling.

We performed all further proteomics sample preparation and data processing steps essentially as previously described[29,61,62]. Briefly, we used urea digestion buffer to solubilize the protein, then reduced (dithiothreitol), alkylated (iodoacetamide), diluted with four volumes of 50 mM ammonium bicarbonate, digested (Lys-C for 3 h, then trypsin overnight), and desalted peptides using C18 STAGE tips made in-house. We suspended the desalted, dried peptides in Buffer A, then estimated peptide concentration by the absorbance at 280 nanometers (Nanodrop). For each sample, 2 µg of peptides were injected into the chromatography system (Easy-nLC 1000, Thermo), which was directly coupled to a Bruker Impact II time-of-flight mass spectrometer, as a single shot unfractionated sample.

**Mass spectrometry data processing**. We searched the mass spectrometry data using MaxQuant (v1.6.8.0). All samples for the queen survey were searched together (123 data files; two samples of the 125 depicted in Fig. 1 were compromised during handling), ensuring a global identification false discovery rate of 1% for both proteins and peptides. We used default search settings, except that label-free quantification was enabled, the minimum number of peptide ratios for quantification was set to 1, and match between runs was enabled. The FASTA database for the search was the newest *A. mellifera* proteome available at the National Center for Biotechnology Information (HAv3.1) along with all honey bee virus and *Nosema* sequences.

We performed differential protein expression analysis using the limma package for R. First, we removed all protein groups that were reverse hits, contaminants, or only identified by site, followed by proteins identified in fewer than 10 samples. Label-free quantification intensities were log2 transformed prior to analysis (available in Supplementary Data 2). The statistical models evaluating protein correlations with sperm viability also included sperm counts, ovary mass, and queen status (failed, healthy, or imported) as fixed effects and source (producer) as a random effect. P values were adjusted using the Benjamini-Hochberg method (10% false discovery rate). The limma output for viability correlations is available in Supplementary Data 3. We verified that all five significant proteins did not correlate with DWV, SBV, BQCV, nor total viral load using a linear least squares model, including sperm viability, status (levels: healthy, failed, imported), and the viral variables as fixed factors.

**Protein-protein correlations and GO enrichments**. We computed the Pearson protein-protein correlation coefficients and set a cut-off of 666 clusters for

hierarchical clustering. This yielded 79 protein clusters containing three or more proteins (clustering results are available in Supplementary Data 4, and clusters containing OBP14, Serpin 88Ea, Lysozyme, HSP10, and Artichoke, specifically, are in Supplementary Data 5). GO terms were then retrieved using BLAST2GO (v4.0), which yielded 1,773 of 1,999 proteins with GO terms (the GO term association table is available in Supplementary Data 6). To identify significantly enriched GO terms within clusters, we performed an over-representation analysis using ErmineJ[63], where proteins within each cluster were considered the 'hit list' and the quantified proteome (1,999 proteins) as the background. To find GO terms that were significantly enriched among proteins correlating with sperm viability, we used the gene-score resampling approach, also using ErmineJ, which we have employed previously[61,62]. This approach utilizes p values as a continuous variable, and detects GO terms that are over-represented among proteins with low p values. Further information can be found at https://erminej.msl.ubc.ca/help/tutorials/running-an-analysis-resampling/. In both types of enrichment analyses, within-test multiple hypothesis testing was corrected to 10% FDR using the Benjamini-Hochberg method.

**Viral analysis**. For viral analysis, $N = 106$ queen heads were shipped to the National Bee Diagnostic Center at Grand Prairie Regional College for analysis. Viral copy numbers were analyzed by RT-qPCR according to the ΔΔCt method. We first shipped a subset of queen heads ($N = 45$ queens) to the diagnostic center on dry ice for analysis of DWV, SBV, BQCV, Israeli acute paralysis virus, Kashmir bee virus, and acute bee paralysis virus. None of the latter three were detectable in the samples; therefore, we submitted a further $N = 61$ samples for analysis of only DWV, SBV, and BQCV.

Queen heads were homogenized in 300 µL of guanidinium isothiocyanate extraction buffer[64]. An aliquot of 200 µL was used to isolate total RNA using the NucleoSpin®RNA kit following manufacturer instructions (Macherey-Nagel Gmbh & Co. KG, Düren, Germany). cDNA was synthesized from 800 ng of total RNA for 20 min at 46 °C in a final volume of 20 µL using the iScript cDNA synthesis kit (Bio-Rad Laboratories, Hercules, USA). cDNA was diluted with 60 µL of molecular biology grade water to a total of 80 µL from which 3 µL were used for qPCR quantification.

Quantification of BQCV, DWV, and SBV infection levels was determined by real-time PCR using primers (Supplementary Data 7) and SSoAdvanced™ Universal SYBR® Green Supermix (Bio-Rad Laboratories, Hercules, USA). Amplification assays were performed by triplicate employing ~30 ng of cDNA in a CFX384 Touch™ Real-Time Detection System (Bio-Rad Laboratories, Hercules, USA). RP49 was chosen as a reference gene. Standard curves were prepared from plasmids harboring the target amplicons with copy numbers diluted from 107 to 102. PCR conditions were 3 min at 95 °C for initial denaturation/enzyme activation followed by 40 cycles of 10 s at 95 °C and 30 s at 60 °C. Specificity was checked by performing a melt-curve analysis 65–95 °C increments of 0.5 °C 2 s/step. Results were analyzed with the CFX Manager™ Software and exported to an Excel spreadsheet to calculate copy numbers.

We used R to perform all statistical analyses. The sperm viability, sperm count, ovary mass, and viral data were first evaluated for normality and equal variance using a Shapiro and Levene test, respectively (viral data were first log-transformed to bring copy numbers to a sensible scale, using $x = \log_{10}(\text{copy number} + 1)$). If the data passed both tests, or failed for normality and passed for equal variance, a classical least-squares linear model was used. If the data failed for equal variance, a weighted least-squares analysis, using the inverse of the fitted data from a first pass unweighted model as the weights. Queen status and queen producer were included as fixed effects in viability, count, and ovary mass statistical models. For ovary masses, differences between failed, healthy, and imported queens were tested with and without producer as a fixed effect, and results from both models are reported. For the viral analysis, we performed statistical tests for each virus separately using queen status (levels: failed, healthy, imported) and sperm viability as fixed factors, as well as the combined viral load (copy numbers from all three viruses were summed prior to log transformation). We were not able to include queen producer as a fixed effect for the viral analysis due to singularities, so we evaluated the effect of producer using a separate statistical model.

**Statistics and reproducibility**. Queens were assigned to failed and healthy groups based on beekeepers' evaluations. Apiary effects and queen status effects were controlled in the proteomics analysis by including queen status as a fixed effect and apiary as a random variable. The finding that failed queens have lower sperm viability than healthy queens replicates the findings of Pettis et al[65]. Given the scale of this queen survey, the survey could not be reasonably replicated with the resources available. We opted to conduct one study with a very large sample size rather than smaller surveys with less statistical power. No sample size calculation was performed, but the number of queens we analyzed ($N = 125$ for phenotypic data, $N = 123$ for proteomics, $N = 106$ for viral analysis) is the largest queen sample size we are aware of for this type of analysis. In all relevant instances, statistical tests were performed in a two-tailed manner. Investigators were not blinded during this work.

**Reporting summary**. Further information on research design is available in the Nature Research Reporting Summary linked to this article.

## Data availability

All novel proteomics raw data, search results, and search parameters are available on MassIVE (www.massive.ucsd.edu, accession MSV000085428). Figures associated with these data are Figs. 2a,b,d, 3, 4a-c,f-h. Associated data are in Supplementary Data 2 and Supplementary Data 3. The mass spectrometry data comparing virgins, mated queens, and drone semen has been previously published[29] and is publicly available at www.proteomexchange.org (accession: PXD013728). Sample metadata underlying Fig. 1 are available in Supplementary Data 1. Global protein abundances and p values for the correlation between sperm viability and spermatheca protein expression are available in Supplementary Data 2. Hierarchical clustering results of protein-protein correlation coefficients are available in Supplementary Data 4. Any other data that support the findings of this study are available from the corresponding author on request.

## Code availability

R code for statistical analyses and figure generation will be provided upon request to the corresponding author, without restriction. R version 3.5.1. Previous releases of R are available at https://cran.r-project.org/bin/windows/base/old/.

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

## Acknowledgements

We would like to acknowledge Kettle Valley Queens, Nicola Valley Honey, Wild Antho, Campbells Gold Honey, Heather Meadows Honey Farm, Six Legs Good Apiaries, Wildwood Queens, Cariboo Honey, and Worker Bee Honey Company for donating failed and healthy queens for this research, and Jordan Tam for assisting with the BLAST2GO analysis. We also thank Patricia Wolf Veiga at the National Bee Diagnostic Center for performing the viral quantification. This work was supported by the Natural Sciences and Engineering Research Council of Canada Discovery grant no. 311654-11 and grants from Genome Canada and Genome British Columbia awarded to L.J.F.; a Project Apis m grant awarded to A.M. and L.J.F, a grant from the Boone Hodgson Wilkinson Trust to A.M. and L.J.F., a Canadian Bee Research Fund grant to A.C. and L.J.F., and a USDA-NIFA grant no. 2016-07962 awarded to J.S.P. and D.R.T.

## Author contributions

A.M. conceived the research plan, performed statistical analyses, generated the figures, wrote the first draft of the manuscript, and handled the revisions. A.C. and A.M. processed the queen samples. A.C. organized the viral analysis. Grants to all authors funded the research. A.C., J.S.P, D.R.T., and L.J.F. helped write and revise the manuscript.

## Competing interests

The authors declare the following competing interests: J.S.P. owns a honey bee consulting business.
