## [Peer Review file · Communications Biology]

Reviewers' comments:

Reviewer #1 (Remarks to the Author):

This interesting study by Alison McAfee and colleagues uses proteomics to investigate the possible trade-off in honeybee queens between sperm maintenance in storage and immune function. This is an important issue because such a trade-off has long been hypothesized, but is less commonly demonstrated. Investigating this issue in honeybees is particularly interesting, given the long periods of time during which queens store sperm. In the article, the authors first show that failed queens have significantly reduced levels of both viable sperm and total sperm in storage. Then, they use quantitative proteomics to investigate levels of spermathecal proteins. After appropriate corrections for multiple tests, they identify five proteins that have significant correlations with sperm viability. Further correlational analyses show that many of these proteins are likely to have roles in regulating some aspects of immunity, providing evidence consistent with a reproduction/immunity trade-off. Finally, they creatively analyze their MS data to ask whether one of their correlated proteins, a serine protease inhibitor, is actively carrying out a canonical type of proteolysis inhibition. Altogether, these data are interesting, well described, and useful for the fields of insect reproduction and immunity, so I think the study will make an appropriate contribution to Communications Biology after addressing some fairly minor points.

1. The number of queens included in the study is in places listed as 123, and elsewhere as 125 (compare, for example, line 30 in the abstract with the sample size given in Fig. 1a, or lines 75 and 79). Please check throughout the manuscript for consistency on this point.
2. In the regression analysis in Fig. 1 and Table 1, did the authors include variables in their linear models that allowed them to control for potential effects of the breeder or the geographic location from which each individual queen came? This is not mentioned in the Methods (lines 330-335), whereas such additional variables are controlled for in later analyses (e.g., lines 362-363). Is there a concern that the 123 (or 125) queens are not truly independent data points, since subsets of them come from the same breeder/area (and thus may have higher levels of genetic similarity)?
3. The use of MS data to test for whether Serpin 88Ea is a canonically active protease inhibitor is neat, and I think this could be a useful example for many researchers studying the proteomics of reproduction, since proteases and protease inhibitors are abundant classes of both female and male reproductive proteins across many taxa. However, the experiment as presented lacks a positive control – that is, the authors have not demonstrated that this method is able to detect evidence of a drop off in the relative quantity of the peptide of a Serpin that covalently binds to a protease. If other serpins (that are predicted to be catalytically active) were also identified in the set of spermathecal proteins, would it be possible to repeat this analysis on one of them to see if that protein's "site peptide" showed a reduced abundance? (I would only suggest this to be done with existing data if possible – no need for additional MS experiments.)
4. In line 44, male-male sperm competition is cited as an example of sexual conflict. Since sexual conflict usually refers to conflict between males and females, though, I suggest re-wording this sentence (perhaps sperm competition could be described as sexual selection instead)?
5. The mention and description of JH in the abstract (line 38) comes a bit out of nowhere; readers familiar with insect immunity will no doubt recognize it, but as worded it is not accessible for others.

6. At lines 129-132, it might be useful to more explicitly spell out a hypothesis for how OBP14 could regulate JH. Are you thinking that OBP14 may bind JH directly, such that higher levels of OBP14 lead to less availability of JH, leading to less immune repression and thus lower sperm levels? While such a model would of course be tested from the direct biochemical study of OBP14 and JH, this kind of experiment is beyond the scope of the current work, so I simply suggest that you unpack the reasoning a bit more here.

7. In Fig. 4d, the first line of the Dmel SPZ protein sequence (TQH...) starts one column before the first line of the Amel SPZ (HQN...), making it unclear whether the sequences are aligned as intended. Please check. Also consider moving this part of the figure to supplemental material. Finally, please define the acronym RCL (for part e of the figure) in the legend.

8. At line 260, it may be useful to note for readers that the third residue of the peptide, R, is not expected to be cleaved by trypsin because it is followed by a P. (There seems to be some debate in the MS literature about how frequently such a cleavage occurs, but since the peptide with the internal R was obviously robustly detected in the authors' data set, that debate is moot here.)

9. In the Methods paragraph that describes the Queens (lines 300-310), please add a bit of information describing the imported queens.

Reviewer #2 (Remarks to the Author):

This study aims to test the hypothesis that a trade-off exists between the reproductive health of honeybee queens (measured as ovary mass, sperm viability and sperm count), and individual immunity (assessed by proteome analysis of the spermathecal contents).

The manuscript is generally well-written, with the exception maybe of some slightly unscientific expressions (such as the use of the term "ideology" for a well-founded scientific hypothesis). The introduction is clearly structured, and the literature review it contains explains very well why a tradeoff between immunity and sperm storage may exist and is worth investigating. However I am afraid that the study presents some methodological shortcomings that may at least partly put into question the validity of the conclusions drawn.

My biggest concern is about the composition of the sample of queens. The three groups used, healthy, "failing" and imported (presumably also healthy?), were taken from different stocks and producers, so that a bias linked to genetic background or rearing conditions cannot be excluded. If, for instance, the two breeders that furnished the majority of the "failed" queens would have used lines of bees genetically overexpressing OBP14, then the over-representation of these two queen origins in the "failed" groups could have led to the false conclusion of a link between OBP14 and sperm storage. Another problem is that the original purpose of the sampling was apparently linked to another study, which is already published and involved exposing queens to heat stress. As heat stress can reduce sperm viability and can also influence the expression of stress- and immunity-related genes, the authors should at least explain why they think that the dataset is still valid with regard to the question of immunity-fertility-trade-offs.

Another problem concerns the strong reliance on 2nd-order correlations (correlations between the proteins found to be linked to sperm viability and other proteins, which themselves were not linked to the primary research question) in the interpretation of the dataset. Interpretation of direct correlations between proteins and biological phenomena can be treacherous enough, so I think that

his part of the discussion, although very well researched, should at least be greatly shortened. Finally, I wonder whether the interpretation of correlations between immune-related genes and sperm storage parameters as evidence for a trade-off is unequivocal. Supposing that parts of the queens had been exposed to pathogens, then this would have influenced not only the expression of immune genes, but potentially also sperm counts and viability. In this case, the correlations between these parameters would stem from a common cause, not a trade-off. As far as I understood, the health status of queens was not checked.

Therefore, although the main hypothesis and many of the proposed interpretations of the data are really interesting, I am afraid that I cannot recommend the publication of this manuscript.

L23: This is only true of some social insects, whereas e.g. termites form stable couples to keep paternity constant over time. I therefore suggest to change this to e.g. “many social hymenoptera”

L37: I do not understand the meaning of “as individuals” here – do you mean that they are positively correlated to lysozyme but not negatively correlated to sperm viability? In this case, it may be worth mentioning them in the discussion, but conclusions drawn from this “2nd degree correlation” may be too weak to be in the abstract.

L42-43: please change to “some hymenopteran queens”

L48: please cite some examples of studies that use HB queens for elucidating the kinds of questions you evoke.

L50: I strongly oppose the use of the term “ideology” in a scientific publication, at least in the natural sciences – an ideal is something you aspire to because you think it SHOULD become reality. A THEORY is what you turn to in order to explain what IS reality.

L64: given that the spermatheca is surrounded by a dense net of tracheoles, I find it hardly difficult to believe that conditions inside should be anoxic – are you certain of this?

L74: English is not my native language but for me, the term “dogma” sounds like something on which you base your faith, not your analytical understanding of how the world works.

L79: here you speak about 125, above (intro) you write about 123 – how is this difference explained?

L80-81: the names for your groups of queens seem slightly ambiguous, as I assume that most of the imported queens were also healthy. What was the reasoning behind including imported queens?

L86: ovary mass is strongly dependent on egg-laying activity during the period directly preceding the moment of sampling, so an important information here would be whether all queens had been taken directly out of the brood nest, or whether some were caged/transported for >2h before dissection.

L93: Although there is of course no strict rule as to when a null hypothesis can be safely seen as confirmed, the convention in the present case would be to presume equality of variances (and, therefore, use methods like ANOVA) only if $P > 0.10$.

L98: I agree, but then why do you report these data if you know they resulted from an artefact (and are therefore meaningless for the verification of your hypotheses)?

L117-118: from the introduction it does not become clear that and why you tested unmated vs. mated queens, so these results come as a surprise to the reader – please include this point in the introduction.

L130 and following: At first I found the hypothesis that OBPs should be involved in JH signalling not very convincing, because you only base your assumption on the reported chemical properties of the (unknown) OBP14 ligand. I therefore googled “juvenile hormone” and “odorant binding” and came across the following article, which directly links OBPs to JH signalling:

<https://www.ncbi.nlm.nih.gov/pmc/articles/PMC5602393/> - maybe you would like to cite it.

Nevertheless, the hypothesis that JH influences sperm storage via immune effects of JH seems a little far-reaching to me – I would think it at least equally likely that JH is regulating sperm storage directly, without the intermediary of any immune effects. After all, the main conserved function of JH in adult insects is in reproduction, not immunity.

Figure 2a: looking at these graphs I find it hard to believe that after correcting for multiple testing (nearly 2.000 proteins!), these correlations should be significant. If I understand right you used Benjamini-Hochberg adjustment with FDR of 10% – is this justifiable?

Figure 2: the title of the figure says that it depicts correlations but in the caption to the figure you describe a regression model. Correlations and regressions are two different things, one used for describing a mere association of variables, one being used for predictions. As your aim is not to predict sperm survival based on protein concentrations, I think you should stick to correlations. I think your reason for using regression might be that you wanted to remove the effects of fixed and variable factors. Maybe use partial correlations for this?

Figure 2: while the figure title speaks of correlations with sperm viability, part c of the figure has nothing to do with sperm viability, only with the gender and mating status of the animals – maybe present 2c as a separate figure?

L186: this is an important point – please give some examples that should fall within these GO terms

L204-205: given that you are talking of proteins whose expression is only correlated to that of other proteins which in turn are directly correlated with sperm viability, this appears like an overstatement – I would suggest to use more moderate terms.

L308: was the time elapsing between removal of queens from their colonies and dissection similar for all queens, or at least similar for the different groups of queens (healthy, failed, imported)?

L300-305: You state that failed and fertile queens were partly not from the same stock. Given that sperm viability/number in failed queens is certainly lower on average, how can you exclude that correlations between these parameters and protein concentrations are not based on the fact that certain proteins were just more strongly expressed in queens from certain origins?

L300-305: Given that diseases would likely affect both the immune status and the reproductive health of queens, I suppose that the presence of diseased queens in your sample would have led to an apparent association between proteins involved in immunity and sperm viability/sperm counts. How did you exclude that any of the queens in the sample carried diseases?

Figure 4: Why is Spaetzle shown here? Just because its expression is correlated to that of L336 and after: I am not an expert of proteomics and will therefore not comment on this part of the methodology.

L304: you state that the data you used is from another published study of yours which involved the exposure of queens to heat stress before dissection – can you exclude that that heat stress has influenced protein expression?

My recommendation for the authors would be to re-interpret their dataset with regard to the more applied question of what may explain queen failure - in this way, the problems with the appropriatedness of the queen sample would partly dissolve.

Reviewer #3 (Remarks to the Author):

This study used proteomics to examine sperm viability and protein trade-off. In general, it is a well written manuscript, there are some interesting points, and the analyses seems appropriate. There are some specific issues that needs to be addressed.

1. Age issues. The author do indicate that the age of the failed queens are unknown (This was a concern I noted earlier in the manuscript). This could be a major issue as 4/5 proteins show that higher levels correlate with reduced sperm viability. Thus, protein accumulation could occur over prolonged periods, so higher levels correlated with age (and likely reduced sperm quality). Age specific analyses would greatly improve this paper.

2. The link between immunity and reproductive trade-offs are not necessarily that strong. Lysozymes have other functions beyond immunity. Also, the immune aspect could be occurring in response to materials released with sperm death. Thus, the increase in lysozyme and other immune factors may only be occurring in direct response to decreasing sperm viability and death rather than the cause of it.

3. Do sperm die and breakdown in the spermathecae? As cell death occurs, proteins are spilled into the local fluid. Is there anything known about the protein content of bee sperm?

4. Are sperm concentrated in the spermathecae? The increase in females could be due to more sperm per volume.

5. Replicates, sample sizes, specific statistics, etc. aren't described well. As an example, were multiple proteomic samples conducted for each bee or only one?

Reviewer #1 (Remarks to the Author):

This interesting study by Alison McAfee and colleagues uses proteomics to investigate the possible trade-off in honeybee queens between sperm maintenance in storage and immune function. This is an important issue because such a trade-off has long been hypothesized, but is less commonly demonstrated. Investigating this issue in honeybees is particularly interesting, given the long periods of time during which queens store sperm. In the article, the authors first show that failed queens have significantly reduced levels of both viable sperm and total sperm in storage. Then, they use quantitative proteomics to investigate levels of spermathecal proteins. After appropriate corrections for multiple tests, they identify five proteins that have significant correlations with sperm viability. Further correlational analyses show that many of these proteins are likely to have roles in regulating some aspects of immunity, providing evidence consistent with a reproduction/immunity trade-off. Finally, they creatively analyze their MS data to ask whether one of their correlated proteins, a serine protease inhibitor, is actively carrying out a canonical type of proteolysis inhibition. Altogether, these data are interesting, well described, and useful for the fields of insect reproduction and immunity, so I think the study will make an appropriate contribution to Communications Biology after addressing some fairly minor points.

Thank you for the thoughtful feedback. We hope we have addressed all your concerns sufficiently, as outlined below.

1. The number of queens included in the study is in places listed as 123, and elsewhere as 125 (compare, for example, line 30 in the abstract with the sample size given in Fig. 1a, or lines 75 and 79). Please check throughout the manuscript for consistency on this point.

Thanks for pointing this out. While 125 queens were assessed for sperm metrics and ovary sizes, only 123 contributed proteomics data owing to unfortunate sample loss during handling (cracked tubes). We have clarified this in the legend of Fig 1 and added a note about this under “Proteomics sample preparation” within the methods (added the text (line 405): “*The final sample count for proteomics was 123 out of 125 initial queens owing to sample losses during handling*”). We also removed reference to our initial sample of 138 queens for clarity (the first batch was later excluded because sperm cells were not removed prior to freezing).

2. In the regression analysis in Fig. 1 and Table 1, did the authors include variables in their linear models that allowed them to control for potential effects of the breeder or the geographic location from which each individual queen came? This is not mentioned in the Methods (lines 330-335), whereas such additional variables are controlled for in later analyses (e.g., lines 362-363). Is there a concern that the 123 (or 125) queens are not truly independent data points, since subsets of them come from the same breeder/area (and thus may have higher levels of genetic similarity)?

We agree that the producer/source should be included as a fixed effect. We had previously not included this in the model because preliminary inspection of producer effects among the BC donors yielded no significant differences. However, we concur that it is best to actually include producer as a fixed effect in the final model. We have

updated the summary statistics in Table 1 and the corresponding figures to reflect the new p values associated with this approach. Overall, the contrasts that were previously significant are still significant, though the p values are not quite as small. The exception to this is for the ovary data, for which the imported queens previously had significantly smaller ovaries. Since Producers “California” and “Kona” obviously completely confound with Imports, and these are the sources with the smallest ovaries, this difference is no longer detectable with the current model. We therefore report both outcomes from the two different models as far as ovary data is concerned, as we still think the follow-up experiments showing the rebound of ovary size after banking is useful to communicate.

Text additions are as follows:

We have added the text (line 78) *“We also included queen producer as a fixed effect to account for potential genetic or environmental differences between sources” to the relevant section in the results. For the ovaries, we explained that differences between imports and the other groups are present but only when producer is not included in the model (line 85): “These differences are not detectable when “producer” is included as a fixed effect in the statistical model, since all the imported queens were produced by California and Kona suppliers.”), and in the Methods, we explain (line 391): “Queen status and queen producer were included as a fixed effects in viability, count, and ovary mass statistical models. For ovary masses, differences between failed, healthy, and imported queens was tested with and without producer as a fixed effect, and results from both models are reported.”*

3. The use of MS data to test for whether Serpin 88Ea is a canonically active protease inhibitor is neat, and I think this could be a useful example for many researchers studying the proteomics of reproduction, since proteases and protease inhibitors are abundant classes of both female and male reproductive proteins across many taxa. However, the experiment as presented lacks a positive control – that is, the authors have not demonstrated that this method is able to detect evidence of a drop off in the relative quantity of the peptide of a Serpin that covalently binds to a protease. If other serpins (that are predicted to be catalytically active) were also identified in the set of spermathecal proteins, would it be possible to repeat this analysis on one of them to see if that protein’s “site peptide” showed a reduced abundance? (I would only suggest this to be done with existing data if possible – no need for additional MS experiments.)

Yes, indeed this analysis is missing a positive control. In the original submission, we opted to conservatively say that we found “No evidence for Serpin 88Ea inhibitory activity” instead of “Serpin 88Ea is not acting as an inhibitor”, to help address this uncertainty. We have considered investigating other serpins as well, but decided against it as we are not aware of an example of a honey bee serpin that, under our experimental conditions, is known to bind appreciable quantities of a protease. Based on serpin biology learned in other systems, we certainly expect that they would, but this knowledge has simply not been developed enough in bees. In short, we would run into the same problems of uncertainty with another serpin (e.g. Serpin 27A or antichymotrypsin, which were also identified in the data) as we do with serpin 88Ea (which is also predicted to be catalytically active), unless we are misunderstanding what the reviewer is suggesting? In light of that, we would consider a few options regarding what to do with the present data: 1) We could keep this section as presented but

acknowledge these important caveats and suggest a better way of doing this type of experiment in the future, 2) we could substantially reduce this section, moving the figure to the supplemental material and again making the caveats clear, or 3) we could remove this analysis from the manuscript entirely. We have currently chosen option 1, but are open to the others if the reviewer thinks a different option is best.

We have added the following paragraph, immediately succeeding our discussion of the serpin peptide analysis (line 299): *“However, we acknowledge that this is an imperfect analysis, since we do not have a good positive control serpin (one which, under our experimental conditions, is known to appreciably covalently bind a protease). Therefore, we cannot confirm the degree of site peptide decoupling that we should expect if the serpin is acting as an inhibitor. In the future, we aim to conduct experiments involving spiking spermathecal fluid with increasing doses of a serine protease to confirm the expected concomitant decoupling of the site peptide from alternate peptides for an array of predicted inhibitory serpins.”*

4. In line 44, male-male sperm competition is cited as an example of sexual conflict. Since sexual conflict usually refers to conflict between males and females, though, I suggest rewording this sentence (perhaps sperm competition could be described as sexual selection instead)?

We have now changed this from “sexual conflict between males” to “sexual selection among males.”

5. The mention and description of JH in the abstract (line 38) comes a bit out of nowhere; readers familiar with insect immunity will no doubt recognize it, but as worded it is not accessible for others.

We have actually removed mention of JH in the abstract while editing for length.

6. At lines 129-132, it might be useful to more explicitly spell out a hypothesis for how OBP14 could regulate JH. Are you thinking that OBP14 may bind JH directly, such that higher levels of OBP14 lead to less availability of JH, leading to less immune repression and thus lower sperm levels? While such a model would of course be tested from the direct biochemical study of OBP14 and JH, this kind of experiment is beyond the scope of the current work, so I simply suggest that you unpack the reasoning a bit more here.

Precisely. We have added the following explanation to the relevant section (line 184):

“We reason that if OBP14 were to bind and sequester free JH, JH may be less able to exhibit its immunosuppressive effects, thus lowering sperm viability by tipping the reproduction-immunity trade-off in favour of immunity. Alternatively, the OBP14-JH complex may bind specific receptors and initiate physiological changes through signalling, rather than sequestration. Further experiments will be necessary to determine the specific molecular mechanisms involved.”

7. In Fig. 4d, the first line of the Dmel SPZ protein sequence (TQH...) starts one column before

the first line of the Amel SPZ (HQN...), making it unclear whether the sequence are aligned as intended. Please check. Also consider moving this part of the figure to supplemental material. Finally, please define the acronym RCL (for part e of the figure) in the legend.

Thank you for identifying this error! During figure editing, the top line (but not other lines) was shifted by one character. We have fixed this mistake.

8. At line 260, it may be useful to note for readers that the third residue of the peptide, R, is not expected to be cleaved by trypsin because it is followed by a P. (There seems to be some debate in the MS literature about how frequently such a cleavage occurs, but since the peptide with the internal R was obviously robustly detected in the authors' data set, that debate is moot here.)

Thank you for the suggestion. We have made a note of this in the figure 4 legend, rather than the body text, to make it easier for the reader to reference the peptide sequence being discussed.

9. In the Methods paragraph that describes the Queens (lines 300-310), please add a bit of information describing the imported queens.

Apologies for the oversight. We have added the following text to the Methods (line 354):

“Imported queens were shipped from producers in Hawaii and California to Edmonton, Alberta, then shipped together to Vancouver within hours of arrival. The queens arrived at 10:30 pm and were dissected and analyzed at the University of British Columbia on the following morning.”

Reviewer #2 (Remarks to the Author):

This study aims to test the hypothesis that a trade-off exists between the reproductive health of honeybee queens (measured as ovary mass, sperm viability and sperm count), and individual immunity (assessed by proteome analysis of the spermathecal contents).

The manuscript is generally well-written, with the exception maybe of some slightly unscientific expressions (such as the use of the term “ideology” for a well-founded scientific hypothesis). The introduction is clearly structured, and the literature review it contains explains very well why a tradeoff between immunity and sperm storage may exist and is worth investigating. However I am afraid that the study presents some methodological shortcomings that may at least partly put into question the validity of the conclusions drawn. Therefore, although the main hypothesis and many of the proposed interpretations of the data are really interesting, I am afraid that I cannot recommend the publication of this manuscript.

Thank you for taking the time to provide such a detailed review. We hope we have been able to sufficiently address the concerns you have raised here and clarified our methodology.

1. My biggest concern is about the composition of the sample of queens. The three groups used, healthy, "failing" and imported (presumably also healthy?), were taken from different stocks and producers, so that a bias linked to genetic background or rearing conditions cannot be excluded. If, for instance, the two breeders that furnished the majority of the "failed" queens would have used lines of bees genetically overexpressing OBP14, then the over-representation of these two queen origins in the "failed" groups could have led to the false conclusion of a link between OBP14 and sperm storage.

We wholeheartedly agree with the reviewer on this point, and for the proteomics data we have already accounted for the issue with our statistical approach. First, regarding the data itself, the two breeders that contributed the most failed queens also contributed a substantial number of healthy queens, so the genetic pools associated with those operations are also represented in the healthy group. Second, we included queen producer as a random effect in our proteomics data analysis model; therefore, if any one producer did happen to have queens with, e.g., unusually high OBP14 levels, this effect is already accounted for. We have added a clarification to the results under Proteomics analysis of spermathecal fluid (line 120): "Since queen source (producer) was included as a random effect in our statistical model, these differences are unlikely to be a result of source bias." The model description was already present in the Methods and the legend of Fig 2 previously.

However, as the first reviewer pointed out, we had not previously accounted for source effects in the data underlying Figure 1 (the viability, count, and ovary mass data analysis prior to proteomics). We had not included queen source in this model because a preliminary inspection of these phenotypic data found no significant relationships with the producers; however, we have now formally included queen source in the model and updated the statistics in Table 1 and the manuscript text to reflect this. As expected, overall, the contrasts which were previously significant remain significant, albeit with somewhat larger p values (but still $p < 0.005$). This is consistent with previous studies of ours (e.g., Tarpy et al. 2012; Delaney et al. 2011), where we find far more variation for reproductive phenotypes within queen-rearing operations than we do among them.

2. Another problem is that the original purpose of the sampling was apparently linked to another study, which is already published and involved exposing queens to heat stress. As heat stress can reduce sperm viability and can also influence the expression of stress- and immunity-related genes, the authors should at least explain why they think that the dataset is still valid with regard to the question of immunity-fertility-trade-offs.

We apologize for the misunderstanding here and hope we have clarified the text now. You are correct, the 125 queens collected here are also linked to another study; however, they were never purposefully temperature stressed. Their role in the other study, which is now published in BMC Genomics, was to see if we could identify any of the candidate stress markers we previously identified within a broad sample of queens from actual beekeeping operations in order to determine 1) how much variation the markers present in the population, and 2) see if they were elevated in the failed queens as a first pass at evaluating their utility in the field. In both that study and the present one, we wanted a sample of queens straight from beekeeping operations, without any particular stress administered in the lab. There was no known history of the queens being stressed in one

way or another. We have added clarifying text to the Methods (line 347): *“The queens were part of a regional survey of participating operations: Queens from different operations were handled similarly and not exposed to environmental stressors in the laboratory.”*

In reality, the original purpose of the sampling is actually that of the present manuscript: a broad survey of failed and healthy queens in BC, as well as imported queens, conducted in collaboration with members of the BC Bee Breeders’ Association. The utility of looking for candidate stress signatures in the existing proteomics data was an added benefit, and that paper happened to be published first.

However, the proteomics data underlying Fig 2c and e do include data from virgins, mated queens, and drones that had been exposed (and not exposed) to heat, so perhaps these are the samples the reviewer is referring to. None of the specific proteins depicted in this figure were among those that were significantly affected by heat, so we think it is still appropriate to represent the data in its present form. If you are interested in those proteins that were affected by heat within that data, it is reported in McAfee *et al.* (2020) Nature Sustainability.

3. Another problem concerns the strong reliance on 2nd-order correlations (correlations between the proteins found to be linked to sperm viability and other proteins, which themselves were not linked to the primary research question) in the interpretation of the dataset. Interpretation of direct correlations between proteins and biological phenomena can be treacherous enough, so I think that this part of the discussion, although very well researched, should at least be greatly shortened.

We accept the point that second order correlations can be problematic. We did our best to determine and demonstrate the likelihood of biological relevance of the approach used, but we have reduced this section as requested, by 260 words (mainly the discussion around HSP10). We have also moved Table 3 to the supplementary information (now Supplementary Table S5) to reduce emphasis on it.

We also point out that the beginning of this section references the paper titled “From protein-protein interactions to protein co-expression networks: a new perspective to evaluate large-scale proteomic data,” which provides an overview of the co-expression matrices technique and under what contexts it has been used previously. We have emphasized this more, now, so that it does not sound like we are developing something new and untested, by adding the text (line 221): *“We therefore exploited protein correlation matrices and hierarchical clustering to make further inferences about the proteins’ potential functions based on proximal associations with other proteins – an approach that has been widely used in other systems.”⁴⁷*

4. Finally, I wonder whether the interpretation of correlations between immune-related genes and sperm storage parameters as evidence for a trade-off is unequivocal. Supposing that parts of the queens had been exposed to pathogens, then this would have influenced not only the expression of immune genes, but potentially also sperm counts and viability. In this case, the correlations between these parameters would stem from a common cause, not a trade-off. As far as I understood, the health status of queens was not checked.

This is an interesting point, and in fact, one of the study's coauthors is making this the topic of her doctoral thesis. Through conducting a series of immune challenges on both mated and virgin queens, she hopes to thoroughly answer these questions.

We have indeed checked a subset of these queens for *Nosema* spores using a hemocytometer and light microscopy, and did not detect appreciable quantities (the vast majority had zero spores observed, and the others had very few), so we did not report this. We have, however, completed molecular analysis for viruses on N = 106 queens, 94 of which also have viability and proteomics data. We have updated the manuscript accordingly, including a new figure panel (Figure 1e) depicting abundances of DWV, BQCV, SBV, and total viral load among healthy, failed, and imported queens (see below).

We have also added a new paragraph titled *Viral analysis* within the Results (line 98):

“To assess patterns of viral abundance in the queen cohorts, we measured deformed wing virus (DWV), sacbrood virus (SBV) and black queen cell virus (BQCV) levels using RT-qPCR. We first analyzed a subset of 45 queens for DWV, SBV, BQCV, as well as acute bee paralysis virus (ABPV), Kashmir bee virus (KBV), and Israeli acute paralysis virus (IAPV), but found no detectable levels of the latter three in any of the queens. We therefore analyzed a further 61 queens for only DWV, SBV, and BQCV (in total, n = 44 healthy queens, n = 13 imported queens, and n = 49 failed queens). We found that failed queens had significantly higher copy numbers of SBV and KBV relative to imported queens and healthy queens, but, surprisingly, lower copy numbers of DWV (Figure 1e). Combining copy numbers of all three viruses into a total viral load, failed queens had significantly higher loads overall than healthy queens, but not imported queens (see Table 1 for all associated p values). We also identified significant effects of queen source (producer) for all three viruses, indicating that the apiaries from which the queens came had characteristic viral profiles (DWV: $p = 4.66 \times 10^{-6}$, $F = 5.2$, $df = 11$ and 94 ; SBV: $p = 3.95 \times 10^{-8}$, $F = 6.66$, $df = 11$ and 94 ; BQCV: $p = 0.0359$, $F = 2.01$, $df = 11$ and 94 ; Total load: $p = 2.44 \times 10^{-8}$, $F = 6.83$, $df = 11$ and 94). N = 98 queens had both viability and virus data, for which none of the viruses nor total viral load were dependent on sperm viability (DWV: $p = 0.330$, $t = -0.979$; SBV: $p = 0.424$, $t = 0.802$; BQCV: $p = 0.579$, $t = 1.79$; Total: $p = 0.878$, $t = -0.153$).”

Furthermore, as we describe at line 124, referring to a new supplementary figure S3:

“To be sure, we checked if these five specific proteins were linked to viral copy numbers (individual viruses as well as total load), and found no significant relationships (Supplementary Figure S3).”

Even with these added data, we still emphasize that the potential for interference by natural pathogenic infections is still possible (line 147):

“We cannot exclude that natural infections could be impacting both immune protein expression and quality metrics. The queens did not have appreciable quantities of Nosema spores visible in their intestinal tract, and while DWV, SBV, and BQCV were detectable, these viruses were not linked to expression of the top proteins linked to sperm viability. However, this is not an exhaustive list of potential pathogens. It is also possible that immune proteins could be elevated as a consequence of sperm death, rather than preceding it. However, in other experiments, we have experimentally stressed queens using techniques that are known to reduce stored sperm viability (i.e. heat exposure) and we did not observe elevated levels of any of the significant proteins we identified here.”

Figure S3. No associations between expression of top five proteins and viral factors. $N = 94$ queens had complete sets of viability, proteomics, and viral data. We evaluated relationships for each protein separately using a least squares linear model, including sperm viability, protein expression, queen status (levels: healthy, failed, imported), and viral copy numbers (levels: DWV, SBV, BQCV, and Total load) as fixed effects. No significant associations were identified ($p > 0.05$), except for protein expression with sperm viability, which we already determined previously ($p < 0.005$).

In other instances throughout the manuscript, we have changed our language to emphasize that the data are consistent with the trade-off hypothesis, rather than outright supporting it.

L23: This is only true of some social insects, whereas e.g. termites form stable couples to keep paternity constant over time. I therefore suggest to change this to e.g. “many social hymenoptera”

Thanks for pointing this out. We have changed the text as suggested.

L37: I do not understand the meaning of “as individuals” here – do you mean that they are positively correlated to lysozyme but not negatively correlated to sperm viability? In this case, it may be worth mentioning them in the discussion, but conclusions drawn from this “2nd degree correlation” may be too weak to be in the abstract.

We agree. We have removed this sentence from the abstract.

L42-43: please change to “some hymenopteran queens”

Done.

L48: please cite some examples of studies that use HB queens for elucidating the kinds of questions you evoke.

We actually don't know of a paper that has investigated this trade off specifically in honey bee queens. Rather, we expect that honey bee queens will become a useful model for these kinds of questions based on unpublished data and the fact that they can be easily manipulated (e.g. via instrumental insemination, and activating ovaries in the absence of insemination). We have changed the text to (line 39): “Honey bee queens could serve as an excellent model system to investigate such processes because they are highly amenable to empirical manipulation” to better reflect that this is an expectation rather than the present situation.

L50: I strongly oppose the use of the term “ideology” in a scientific publication, at least in the natural sciences – an ideal is something you aspire to because you think it SHOULD become reality. A THEORY is what you turn to in order to explain what IS reality.

We have changed ideology to ‘hypothesis,’ as we are not sure the reproduction-trade-off explanation has yet reached the strength of a theory.

L64: given that the spermatheca is surrounded by a dense net of tracheoles, I find it hardly difficult to believe that conditions inside should be anoxic – are you certain of this?

Previous investigations have determined that the amount of dissolved oxygen within the spermatheca is very low, according to the cited paper (Paynter et al. Scientific Reports, Figure 4). We are not aware of literature that refutes it and the levels reported would certainly be considered anoxic.

L74: English is not my native language but for me, the term “dogma” sounds like something on which you base your faith, not your analytical understanding of how the world works.

I think this is a language difference. For example, in introductory biology courses, the process of DNA being transcribed to RNA, which is translated to proteins, is ubiquitously referred to as “the central dogma of molecular biology.” It can be thought of as a set of axioms or principles. On that note, however, we think the use of dogma here is actually too strong, and we have changed this to ‘hypothesis.’

L79: here you speak about 125, above (intro) you write about 123 – how is this difference explained?

Thanks for pointing this out. While 125 queens were assessed for sperm metrics and ovary sizes, only 123 contributed proteomics data owing to unfortunate sample loss during handling (cracked tubes). We have clarified this in the legend of Fig 1 and added a note about this under “Proteomics sample preparation” within the methods (added the text (line 423): “The final sample count for proteomics was 123 out of 125 initial queens owing to sample losses during handling”). We also removed reference to our initial sample of 138 queens for clarity (the first batch was later excluded because sperm cells were not removed prior to freezing).

L80-81: the names for your groups of queens seem slightly ambiguous, as I assume that most of the imported queens were also healthy. What was the reasoning behind including imported queens?

The reasoning for including imported queens was because the BC Bee Breeders’ Association was initially interested in testing if local queens were of similar or different quality to the queens most commonly imported to the region. The imported queens are presumably healthy, but with the speed of queen production and harvesting, as well as risk of temperature stress as air cargo, we cannot be as certain of their health status as we can with careful local beekeepers. The data show that imported and healthy local queens have essentially the same quality metrics, which is one of the results we wish to communicate to the industry.

L86: ovary mass is strongly dependent on egg laying activity during the period directly preceding the moment of sampling, so an important information here would be whether all queens had been taken directly out of the brood nest, or whether some were caged/transported for >2h before dissection.

We agree, and have added that ovary mass can be influenced by caging time and worker care at line 88. At line 82 we also begin to address this:

“Ovary masses also differed significantly between groups – imported queens had significantly lower ovary masses compared to either healthy queens ($p = 0.0021$, $t = 3.1$, $df = 134$) or to failed queens ($p = 0.038$, $t = 2.1$, $df = 134$), but ovary mass can be strongly influenced by caging time and worker care.”

Further details about queen shipping conditions are already available in the first paragraph of the Methods, around line 352:

“The queens arrived via overnight ground transportation (ACE Courier or via post) to the University of British Columbia in Vancouver and were sacrificed for analysis immediately upon arrival. Imported queens were shipped from producers in Hawaii and California to Edmonton, Alberta, then shipped together to Vancouver within hours of arrival. The queens arrived at 10:30 pm and were dissected and analyzed at the University of British Columbia on the following morning.”

L93: Although there is of course no strict rule as to when a null hypothesis can be safely seen as confirmed, the convention in the present case would be to presume equality of variances (and, therefore, use methods like ANOVA) only if $P > 0.10$.

Exactly. The results of the Levene test show that $P = 0.051$; therefore, we thought it safest to reject the null hypothesis and employ a weighted least-squares regression. Unless we misunderstand what the reviewer is suggesting?

L98: I agree, but then why do you report these data if you know they resulted from an artefact (and are therefore meaningless for the verification of your hypotheses)?

The malleability of ovary size seems to be something observed by people frequently doing detailed evaluations of queens, but it does not seem to be a reported phenomenon. I am frequently asked this question by beekeepers, that is, if caging or shipping affects queen mass or ovary size (e.g. “Why do imported queens appear so much smaller than my home-grown queens?”). Therefore, we chose to include these data to help clarify what is going on for beekeepers.

L117-118: from the introduction it does not become clear that and why you tested unmated vs. mated queens, so these results come as a surprise to the reader – please include this point in the introduction.

We have added the following sentence to the last paragraph of the introduction (line 66):
“Additionally, there is limited data on how proteins linked to sperm viability change after mating, when the queen must transition from storing no sperm to maintaining sperm viability for years.”

L130 and following: At first I found the hypothesis that OBPs should be involved in JH signalling not very convincing, because you only base your assumption on the reported chemical properties of the (unknown) OBP14 ligand. I therefore googled “juvenile hormone” and “odorant binding” and came across the following article, which directly links OBPs to JH signalling: <https://www.ncbi.nlm.nih.gov/pmc/articles/PMC5602393/> - maybe you would like to cite it. Nevertheless, the hypothesis that JH influences sperm storage via immune effects of JH seems a little far-reaching to me – I would think it at least equally likely that JH is regulating sperm storage directly, without the intermediary of any immune effects. After all, the main conserved function of JH in adult insects is in reproduction, not immunity.

Thank you very much for pointing out this article on JH binding in mosquitoes. It was published around the time we were preparing the manuscript, so we had not yet seen it and referenced it. It has been added now and is certainly an asset to our argument.

We agree that it is plausible that JH regulates sperm storage directly. We wish to clarify that we do not necessarily think that JH is influencing sperm viability via immune effects. Rather, we see JH as a hub hormone that itself has multiple effects (as has already been demonstrated for other physiological processes). So, we agree that it could be having a direct or indirect influence on sperm viability, which we have added to line 181:

“We thus speculate that in the spermathecal fluid, OBP14 may be involved in hormonal signalling that regulates queen immunity, and OBP14-mediated JH signalling may influence sperm viability directly or indirectly through immune effects. We reason that if OBP14 were to bind and sequester free JH, JH may be less able to exhibit its immunosuppressive effects, thus lowering sperm viability by tipping the reproduction-immunity trade-off in favour of immunity. Alternatively, the proposed OBP14-JH complex may bind specific receptors and initiate physiological changes through signalling, rather than sequestration. Further experiments will be necessary to determine the specific molecular mechanisms involved.”

Figure 2a: looking at these graphs I find it hard to believe that after correcting for multiple testing (nearly 2,000 proteins!), these correlations should be significant. If I understand right you used Benjamini-Hochberg adjustment with FDR of 10% – is this justifiable?

The correlations marginally passed the threshold of 10% FDR, and when the threshold is set to 5%, as is more often used, only OBP14 is considered significant, which is also the protein we are most interested in functionally. In our experience with high-throughput data, a 10% FDR threshold is certainly on the higher end of what is acceptable, whereas 5% is more conventional. At a 10% FDR, the chances of all the five proteins we discuss are false discoveries is 1 in 100,000, which seems like an appropriate level of certainty to warrant discussing even those which do not pass the 5% cut off. It’s worth noting that a 5% threshold is conventional for highly controlled experiments, e.g. in which a more homogenous population is divided into treatment groups. We think that a widespread survey with queens from multiple locations, genetic sources, and ages, yielding data that is significant, even at the 10% level, is worth discussing.

Figure 2: the title of the figure says that it depicts correlations but in the caption to the figure you describe a regression model. Correlations and regressions are two different things, one used for describing a mere association of variables, one being used for predictions. As your aim is not to predict sperm survival based on protein concentrations, I think you should stick to correlations. I think your reason for using regression might be that you wanted to remove the effects of fixed and variable factors. Maybe use partial correlations for this? Figure 2: while the figure title speaks of correlations with sperm viability, part c of the figure has nothing to do with sperm viability, only with the gender and mating status of the animals – maybe present 2c as a separate figure?

We have changed the title of the figure caption to “Proteins associated with sperm viability” to try to improve clarity and to address the disconnect between Fig 2c and the other panels. We believe our statistical approach is the appropriate one, but we mistakenly referred to our linear model as a regression. Our understanding is that linear models are not obligatorily predictive – they are used to model the relationships between, in this case, protein abundance data and the explanatory variables. The approach we used has been used countless times for similarly structured large-scale gene expression data sets, and indeed, this is exactly the kind of dataset the limma package was built for – we simply referred to it incorrectly. We have corrected all instances of this in the manuscript and we hope with the edited title, the links between the panels is more coherent.

L186: this is an important point – please give some examples that should fall within these GO terms

In an effort to reduce this section of the manuscript, as the reviewer suggested earlier, we have actually removed this part of the discussion. However, the most obvious examples that come to mind are Lysozyme – a well known innate immune effector regulated by Toll signalling, which was not assigned GO terms linked to these processes – and Serpin88Ea, which is an activator of Toll in *Drosophila*. The same goes for prophenoloxidase, which was also robustly identified in our data but was not linked to innate immune GO terms.

L204-205: given that you are talking of proteins whose expression is only correlated to that of other proteins which in turn are directly correlated with sperm viability, this appears like an overstatement – I would suggest to use more moderate terms.

We agree that this may have been an overstatement as previously written. We have edited this section down to the text below (line 249):

“While the cluster containing OBP14 did not yield any significant GO terms, two of the other cluster members are Apolipophorin I/II and Hexamerin 70a, both of which are also involved in JH binding in other insects, suggesting that OBP14, Apolipophorin I/II, and Hexamerin 70a could be facilitating hormone trafficking. Others have shown that JH diet supplementation improves sperm viability, and JH serves as an immunosuppressant in mated females of other insects, which is consistent with the reproduction-immunity trade-off hypothesis. Indeed, Kim et al. recently identified a mosquito OBP which binds JH and activates innate immune defenses – a mechanism which, according to the reproduction-immunity trade-off hypothesis, would be consistent with high levels of OBP14 being associated with low sperm viability in our data.”

L308: was the time elapsing between removal of queens from their colonies and dissection similar for all queens, or at least similar for the different groups of queens (healthy, failed, imported)?

Yes, for queens supplied directly by beekeepers, they typically removed the queens from their colonies the day before shipping and then shipped them to the lab via overnight ground transportation, or drove them to the lab themselves. All of the shipments arrived

within the estimated delivery time, but I am aware of at least one exception where the courier failed to pick up a package, delaying the actual shipment by a day. So we are hesitant to describe exact methods in this section, due to there being at least one exception that we are aware of (it is always possible that a beekeeper adjusted their schedule e.g. due to weather and failed to communicate that detail). We expect that the variation in the delivery time would be +/- 1 day out of the colony, for both failed and healthy queens. Imported queens are even less controlled – without direct communication with the people on the ground, pulling the specific queens belonging to these shipments, it is impossible to know exactly how long they were outside their mating nucs.

L300-305: You state that failed and fertile queens were partly not from the same stock. Given that sperm viability/number in failed queens is certainly lower on average, how can you exclude that correlations between these parameters and protein concentrations are not based on the fact that certain proteins were just more strongly expressed in queens from certain origins?

We believe we have addressed this concern already, as best we can, in our response to the first point (major concern) raised by the reviewer regarding the proteomics data. A similarly important point was also raised by Reviewer 1, regarding the viability, count, and ovary mass data (i.e. the phenotypic data, not the proteomics data). Here is our response to that:

We agree that the producer/source should be included as a fixed effect. We had previously not included this in the model because preliminary inspection of producer effects among the BC donors yielded no significant differences. However, we concur that it is best to actually include producer as a fixed effect in the final model. We have updated the summary statistics in Table 1 and the corresponding figures to reflect the new p values associated with this approach. Overall, the contrasts that were previously significant are still significant, though the p values are not quite as small. The exception to this is for the ovary data, for which the imported queens previously had significantly smaller ovaries. Since Producers “California” and “Kona” obviously completely confound with Imports, and these are the sources with the smallest ovaries, this difference is no longer detectable with the current model. We therefore report both outcomes from the two different models as far as ovary data is concerned, as we still think the follow-up experiments showing the rebound of ovary size after banking is useful to communicate.

L300-305: Given that diseases would likely affect both the immune status and the reproductive health of queens, I suppose that the presence of diseased queens in your sample would have led to an apparent association between proteins involved in immunity and sperm viability/sperm counts. How did you exclude that any of the queens in the sample carried diseases?

This is indeed an important question, and we have addressed this concern extensively with added data as described under major concern #4 above.

Figure 4: Why is Spaetzle shown here? Just because its expression is correlated to that of

The remainder of this comment appears to be cut off. We simply included spaetzle to complete the SPZ-SPE-Serpin triangle. We can remove it, if deemed unnecessary or confusing.

L336 and after: I am not an expert of proteomics and will therefore not comment on this part of the methodology.

Reviewer 1 has expertise in proteomics and mass spectrometry, and has offered some useful feedback for this section.

L304: you state that the data you used is from another published study of yours which involved the exposure of queens to heat stress before dissection – can you exclude that that heat stress has influenced protein expression? My recommendation for the authors would be to re-interpret their dataset with regard to the more applied question of what may explain queen failure - in this way, the problems with the appropriateness of the queen sample would partly dissolve.

We hope that we have addressed this concern sufficiently under point number 2 above (within major concerns). Again, we appreciate the detailed review and hope that you will find the revised manuscript improved.

Reviewer #3 (Remarks to the Author):

This study used proteomics to examine sperm viability and protein trade-off. In general, it is a well written manuscript, there are some interesting points, and the analyses seems appropriate. There are some specific issues that needs to be addressed.

1. Age issues. The author do indicate that the age of the failed queens are unknown (This was a concern I noted earlier in the manuscript). This could be a major issue as 4/5 proteins show that higher levels correlate with reduced sperm viability. Thus, protein accumulation could occur over prolonged periods, so higher levels correlated with age (and likely reduced sperm quality). Age specific analyses would greatly improve this paper.

We agree with the reviewer on the point that age-specific analyses would improve this body of work. In general, we expect that age effects have been broadly accounted for with the way we conducted our statistical analysis. Failed queens tended to be older than healthy queens (all of the healthy queens were approximately one month old). The ages of twenty-two of the failed queens (almost half) are unknown, but of those that are defined, the average age is 6.3 months. We did not want to identify proteins associated with sperm viability simply as an artefact of queen age or other extraneous variables linked to colony failure, so we included failed vs. healthy colony status as a fixed effect in our linear model. This means that the resulting proteins associated with sperm viability are indeed still associated when the failed vs. healthy effect (which conveniently confounds with age, since the failed queens are older, on average) is removed. We have added this explanation to the Results and Discussion (line 120):

“Since queen source (producer) was included as a random effect in our statistical model, these differences are unlikely to be a result of source bias. Furthermore, colony health status (‘failed,’ ‘healthy,’ and ‘imported’) was included as a fixed effect in the model, and since queens heading failed colonies also tended to be older and had a higher viral load, these proteins are unlikely to simply be linked to sperm viability indirectly through aging queens or differences in viral titer (queen ages, where known, are listed in Supplementary Table S1).”

As indicated above, we have also added two more columns to the sample metadata table (Supplementary Table S1) – one for queen year, and one for approximate age in months, if known. We are currently conducting an ongoing follow-up queen survey, working closely with a few local beekeepers, to collect failed and healthy queens with known ages in order to address this point. We have so far collected around fifty queens with known ages and known colony health status, and we aim to use these as a validation data set.

However, it is not obvious to us that age is necessarily associated with protein accumulation. In fact, looking at the rest of the proteomics data (Supplementary Table S4), we see that the slope of the correlation between the other proteins and sperm viability is relatively evenly split between positive and negative associations. In fact, it is slightly biased toward a positive relationship, with 798 proteins being negatively associated and 1198 being positively associated. It just happens that four out of the five significant proteins are negatively correlated with viability – this does not reflect the overall trend in the data.

2. The link between immunity and reproductive trade-offs are not necessarily that strong. Lysozymes have other functions beyond immunity. Also, the immune aspect could be occurring in response to materials released with sperm death. Thus, the increase in lysozyme and other immune factors may only be occurring in direct response to decreasing sperm viability and death rather than the cause of it.

This is a good point that lysozymes have diverse functions. In fact, in mammals, lysozymes are commonly found in testis and sperm (particularly sperm tails). We thought it might be possible that the lysozyme we identified originated from the sperm itself, and could increase in the spermatheca as more sperm died, as the reviewer is suggesting in point number 3 (see our response to that comment below – based on our data, it is unlikely that is occurring).

The reviewer raises an interesting point about sperm death itself potentially causing immune stimulation. We had not previously considered this, but we have another previously published dataset which can help answer this question.

Heat-shock is one method of experimentally reducing stored sperm viability. We recently published a paper where we performed differential proteomics on heat-shocked queens in order to find candidate molecular markers to aid with queen failure diagnostics (see McAfee et al. (2020) Nature Sustainability). Neither lysozyme nor the other proteins that significantly associate with sperm viability were elevated in the stressed queens (which we would expect to see if they are an immune response to increased sperm death). We have added a sentence addressing this around line 121, including another caveat pointed out by one of the other reviewers. We think this wording helps temper our claims (line 147).

“We cannot exclude that natural infections could be impacting both immune protein expression and quality metrics. The queens did not have appreciable quantities of Nosema spores visible in their intestinal tract, and while DWV, SBV, and BQCV were detectable, these viruses were not linked to expression of the top proteins linked to sperm viability. However, this is not an exhaustive list of potential pathogens. It is also possible that immune proteins could be elevated as a consequence of sperm death, rather than preceding it. However, in other experiments, we have experimentally stressed queens using techniques that are known to reduce stored sperm viability (i.e. heat exposure)²⁹ and we did not observe elevated levels of any of the significant proteins we identified here.”

3. Do sperm die and breakdown in the spermathecae? As cell death occurs, proteins are spilled into the local fluid. Is there anything known about the protein content of bee sperm?

Yes, in fact, we have previously performed quantitative proteomics experiments on honey bee drone ejaculates (this is the data that contributed to Figure 2c, comparing abundances in drone ejaculates, virgin queen spermathecae, and mated queen spermathecae). It is precisely to address this point that we investigated protein abundances for our five proteins of interest, including lysozyme, in drone semen as well (as shown in Fig 2c) – because we wanted to get a better idea of the likely origin (sperm or fluid) of the proteins. We did not identify lysozyme in any of the semen samples, so we think it is highly unlikely that dying sperm were the source of this protein.

Of the two proteins of interest that were also identified in the semen samples, one (Serp11 88Ea) was more abundant in the semen than in spermathecal fluid, while the other (OBP14) was significantly less abundant. As the reviewer points out, since both proteins are negatively associated with sperm viability, having any amount within sperm at all could theoretically lead to the associations we observe. However, protamines are among the most abundant sperm nuclear proteins in animals, so we are conveniently able to check if this protein correlates with sperm viability. If what the reviewer suggests is happening is indeed a problem, protamine abundance should be elevated in samples with low sperm viability. We have added supplementary Figure S3a showing that is not the case. While one might expect protamine abundance to be elevated in samples with the highest absolute number of dead sperm (rather than % viability), we show in Figure S3b that is also not the case. We have now added a note indicating this around line 167:

“Although the abundance of OBP14 in semen is low, it is possible that sperm death and subsequent release of proteins could contribute to the abundance patterns we observe (the same is true for Serpin 88Ea, which is also present in semen). To check this, we correlated protamine-like protein (a highly abundant sperm nuclear protein) with both sperm viability and absolute number of dead sperm, and found no significant correlations (Supplementary Figure S4, Pearson correlation, $p = 0.215$ and $p = 0.321$, respectively).” The protein was also sparsely identified in only 36 out of 123 samples, and was likely a result of sporadic sperm lysis during sample handling. Therefore, we reason that it is unlikely that the release of sperm proteins upon death can explain the negative correlations we observe for OBP14, Serpin 88Ea, Lysozyme, and Artichoke.

Figure S4. Correlations of Protamine-like protein (XP_026294833.1), an abundant sperm nuclear protein, with sperm viability (a) and number of dead sperm (b).

4. Are sperm concentrated in the spermathecae? The increase in females could be due to more sperm per volume.

We believe that the reviewer is referring to Figure 2c or possibly 2e? In any case, we do not expect that sperm concentration to be higher in spermathecae compared to drone ejaculates. There are 7.5-12 million sperm cells in a single drone's seminal vesicles, and somewhat less than that is acquired from a forced ejaculation. We typically acquired about 1 microliter of semen from each drone, and it is reasonable to assume that this represents about 50-75% of the total sample (we cannot collect 100% of the sample in order to avoid collecting the seminal mucous layer). Assuming 75%, this would correspond to 5.6-9 million sperm. The average spermatheca is 1.25 mm, which corresponds to a spherical volume of just over one microliter. While our data show that the spermathecae contains around 3 million sperm, other literature reports approximately 6 million. So, if anything, we actually expect that the concentration of sperm in drone semen is actually the same or higher than in the spermatheca.

5. Replicates, sample sizes, specific statistics, etc. aren't described well. As an example, were multiple proteomic samples conducted for each bee or only one?

Lines 403-416 now describe specific statistics. At line 418-423 in the methods, we have clarified that the protocols as described were conducted on spermathecal supernatants from each queen (i.e. each queen was a separate sample). At the end of the paragraph, we also clarify that the peptides were analyzed as a single-shot, unfractionated injection. Sample sizes are described throughout the Methods, as well as more clearly in the Results and Discussion.

Regarding replicates, sample sizes, and statistics, we suspect that the reviewer received an initial version of the manuscript which indeed did not report these parameters in detail. After the initial submission, we filled out a publication checklist as requested by the editor, and after reviewing the expectations we realized that we were missing this key

information. We immediately updated the manuscript and sent it back to the editor for distribution, but based on the reviewer's comments the version that was sent out was likely from before these updates. The current version has sample sizes, degrees of freedom, F statistics, T statistics, p values, and multiple hypothesis testing thresholds described in each figure legend as well as in the Results and Discussion section.

REVIEWERS' COMMENTS:

Reviewer #1 (Remarks to the Author):

The authors have done a good job addressing the points I raised in my initial review. I appreciate their attention to the statistical analysis issue of queen source. I also appreciate the difficulty of finding a sure-fire positive control for the proteolysis inhibitor proteomic experiment. I think the first option outlined by the authors in their response to reviews (keeping the experiment in the manuscript, and just modifying the text) is sufficient, and that there will be value to other scientists studying proteolysis regulation in reproduction just in seeing this kind of analysis attempted.

Thanks for this interesting manuscript and for your thoughtful responses to the first round of review!

Reviewer #2 (Remarks to the Author):

I found the corrected version of the manuscript greatly improved, particularly with regard to the description of methodology. The authors have also made a great effort to justify their interpretation of correlations as relationships of cause and effect, by excluding some potential common causes of sperm storage- and immune-effects (pathogens, queen age).

I am not satisfied with the justification given for the use of a 10% FDR - for me, the fact that the study was poorly-controlled (maybe poorly controllable) is a reason to mistrust the results rather than trusting them even more (by interpreting correlations as relevant that would not have been considered as such with a conventional FDR of 5%).

Nevertheless, I agree that the manuscript should now be published.

Reviewer #3 (Remarks to the Author):

The authors have addressed my previous concerns.